



# Episodic sediment supply to alluvial fans: implications for fan incision and morphometry

Anya. S. Leenman[1,2] and Brett. C. Eaton[1]

[1]Department of Geography, University of British Columbia, Vancouver, Canada
[2]School of Geography and the Environment, University of Oxford, Oxford, U.K.

**Correspondence:** Anya. S. Leenman (anya.leenman@chch.ox.ac.uk)

**Abstract.** Sediment supply is widely believed to be a key control on alluvial fan morphology and channel dynamics. Although the sediment supply to natural fans is rather episodic, experimental studies of alluvial fans often use constant sediment supply rates, making it difficult to relate fan dynamics to the magnitude and frequency of sediment supply in the field. This paper presents a series of experiments designed to test the impact of episodic sediment supply on fan evolution and dynamics.

We compare four experiments, each with the same mean sediment supply but different durations of high- and low-supply periods. The experiments show that fan morphology and channel dynamics respond systematically to the temporal elongation of sediment supply oscillations: longer supply cycles generate flatter fans with more trenched channels. These results highlight how different basin conditions might generate different fan morphologies: supply limited basins with intermittent sediment supply might generate fans that are flatter than expected. Our results raise the question of whether a constant sediment supply

in experimental models can adequately characterise the dynamics of natural fans in the field.

## 1 Introduction

The shapes of alluvial fans reflect the fluxes of water and sediment from their source basins. Field and experimental studies show that, for a given flow rate, fans become steeper as the sediment supply (and importantly, the sediment concentration) increases (Ashworth et al., 2004; Bryant et al., 1995; Bull, 1964; Delorme et al., 2018; Hooke, 1968b; Whipple et al., 1998).

Fan channel dynamics also change, with more frequent avulsions at higher sediment supply rates (Ashworth et al., 2004; Bryant et al., 1995). This influence of sediment concentration is reflected in the inverse relation between upstream catchment area and fan slope (e.g. Blair and McPherson, 1994, Figure 23). Within-basin sediment storage means that while larger basins generate higher discharges, they typically have lower sediment concentration (often fluvial flows as opposed to debris flows), and hence form lower gradient fans (Blair and McPherson, 1994; Crosta and Frattini, 2004; De Scally and Owens, 2004; Harvey, 1984;

Kostaschuk et al., 1986; Stokes and Gomes, 2020; Tomczyk, 2021).

Despite these established linkages between sediment supply and fan forms, little is known about how oscillations in the sediment supply or concentration govern fan morphology. In the field, fans experience temporal variation in their sediment supply, depending on the frequency and magnitude of erosion events in their source basins (e.g. Cabre et al., 2020; Davies and Korup, 2007; Frechette and Meyer, 2009; Kesel and Lowe, 1987; Meyer and Pierce, 2003; Pierce and Meyer, 2008; Talbot and





Williams, 1979; Wells and Harvey, 1987, among others). The nature and timing of sediment delivery to a fan is moderated in turn by the size of the source basin and the connectivity of sediment transport pathways therein; disconnectivity within larger basins can mean that sediment inputs from erosional events do not make it to the basin outlet to build alluvial fans at all (Stokes and Mather, 2015; Wang et al., 2008).

While the episodic nature of sediment supply to fans is well documented, only a few alluvial fan experiments have explicitly 30  investigated the effects of episodic sediment supply. Schumm et al. (1987, p. 316) conducted a preliminary study of fans built from episodic flood events generated via "a burst of artificial rainfall applied by a hand-held nozzle" to the experimental source basin. Davies and Korup (2007) generated random pulses in their "background" experimental sediment supply, while investigating the impact of large pulses superimposed on this input. Apart from these two studies, most alluvial fan experiments used a constant sediment supply. Moreover, there has not been a systematic experimental study of how episodic sediment supply 35  affects alluvial fan morphology in the long-term, or the stability and patterns of stream channels upon the fan.

This paper presents four alluvial fan experiments designed to address this knowledge gap. The experiments demonstrate the impacts of variable sediment supply on alluvial fan morphology, channel patterns, and change thereof. All the experiments had the same mean sediment supply rate, but varying durations of high-supply and low-supply conditions. These scenarios allow us to investigate the effects of periodic oscillations around the mean sediment supply rate.

Through comparing the four experimental scenarios, this paper considers two key questions:

1.  How do abrupt changes in sediment supply rate influence fan morphology, channel patterns and dynamics?

2.  How does the duration of sediment supply oscillations govern their impact?

The experiments presented here show that abrupt sediment supply oscillations can generate distinctive responses in the fan gradient, channel patterns, lateral mobility and morphologic reworking rates. Moreover, comparisons between the experiments 45  show that fan morphology changes systematically with oscillation duration. While the experiments indicate how fans are likely to respond to changes in sediment concentration during or between floods, they also highlight the distortions that arise from using a constant sediment supply rate to represent systems whose sediment supply is intermittent in the field.

## 2  Methods

This paper describes four experiments conducted in the alluvial fan simulator at the University of British Columbia's Bio-50  geomorphology Experimental Laboratory. The experiments presented here are named Runs 1, 7, 8 and 9. Results from Run 1 have been presented in detail in (Leenman and Eaton, 2021), and compared to another set of fan experiments (Runs 2-4) in (Leenman et al., 2022). The experimental set-up, sediment mixture, scaling approach and data collection system were the same as in those papers, but they are summarised here for completeness.





## 2.1 Model set-up

Our stream table experiments used a physical model of a generic gravel-cobble alluvial fan. The stream table was 2.44 × 2.44 × 0.3 m (Figure 1). We delivered water and sediment to the fan-head through a 0.2 × 0.5 × 0.3 m feeder channel at one corner of the table. Water was input from a constant head tank. Sediment was input from a sediment feeder with a rotating pipe outlet; the sediment supply rate was set by the inclination of the pipe. Sediment and water were first mixed in a funnel, then dropped into the head of the feeder channel. Sediment aggraded and degraded freely in the feeder channel, as we would expect for a

confined reach upstream of a natural fan in the field.

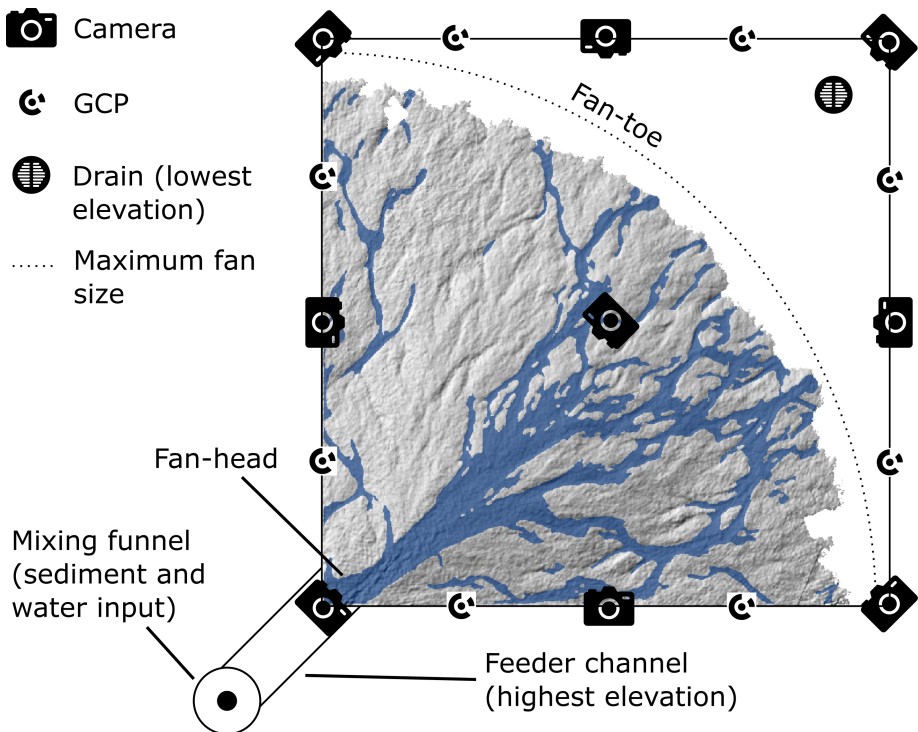

**Figure 1.** Experimental design (not to scale). Water and sediment mix in the funnel and drop into the experiment at the head of the feeder channel, where sediment aggrades and degrades freely. The hillshaded topography and flow map example are from Run 1 repeat 2, 20 hours into the experiment.

We tilted the stream table to 0.0002 m m$^{-1}$ (0.02 %), the minimum slope required to trigger flow toward a drain in one corner and avoid the formation of a "lake" in the experiment. We roughened the experimental boundary with 2 mm sand grains and LEGO® sheets, glued to the table and walls respectively. The water was dyed blue to permit automatic channel extraction from overhead photographs.

We collected data with a system inspired by Structure-from-Motion photogrammetry. Leenman and Eaton (2021) and Leenman (2021) give detailed descriptions of the data collection system and its accuracy, so we present a brief summary here. The



system consisted of an array of nine digital single-lens reflex cameras mounted above the stream table (locations shown in Figure 1). The cameras captured photos synchronously, at one-minute intervals. In experiments with sediment supply changes, the first photo was always ~30 seconds after the change in supply rate. We georeferenced each set of nine photos to a local

coordinate system, using a set of eight ground control points on the table walls. We processed each photo set using Agisoft Photoscan (2018) to generate a topographic point cloud (~280,000 points per m$^2$) and co-registered orthomosaic (1 mm resolution) for each minute of the experiment.

## 2.2    Experimental Approach

Our experimental alluvial fan is a "similarity-of-process" or "analogue" model (c.f. Hooke, 1968a; Paola et al., 2009), as are

most physical models of fans (e.g. Bryant et al., 1995; Clarke et al., 2010; Davies and Korup, 2007; Van Dijk et al., 2009; De Haas et al., 2016, 2018; Hamilton et al., 2013; Hooke, 1967, 1968b; Hooke and Rohrer, 1979; Miller et al., 2019; Piliouras et al., 2017; Reitz and Jerolmack, 2012; Schumm et al., 1987). The key processes in the model (fluvial sediment entrainment, transport and deposition) are similar to those on fans in the field. In alluvial fan experiments, it is challenging to maintain Froude scaling between the model and any field prototype, due to the large scaling ratio that is necessary to build a conveniently

small laboratory model. This lack of Froude scaling means that it is inappropriate to extrapolate rates and volumes measured in the experiment to field settings. Nevertheless, comparisons *between* the different experiments demonstrate how natural fans might respond to different frequencies of sediment delivery. Such comparisons also highlight the distortions introduced through temporal averaging in the experimental inputs.

    The slope of the fan itself, and the dimensions of the channels upon it, were self-formed. Therefore, it was not possible

to control the Froude or Reynolds numbers during the experiments. To give readers an idea of the flow dynamics, we have estimated these parameters for the fan-head (where flow was often confined to one channel) in run 1 (constant inputs). The estimated Froude number ($Fr$) was ~1.9; this value aligns with observations of supercritical flow during floods on fans in the field (Beaumont and Oberlander, 1971; Rahn, 1967). Further down-fan, flow likely became subcritical as it spread into multiple distributaries. The particle Reynolds number ($Re^*$) was estimated (using the $D_{84}$ as a representative grain size) to

be 66. This conforms to the threshold of 15 proposed by Parker (1979) and Ashworth et al. (1994), although it is below the minimum of 70 recommended by Schlichting and Gersten (2016) and Yalin (1971). The estimated Reynolds number ($Re$) was 1200, placing the flow in the transitional regime between laminar and turbulent flow (and preventing the attainment of Froude scaling). Nevertheless, Malverti et al. (2008) showed that even laminar flow can transport sediment at rates which fit the Meyer-Peter and Müller (1948) relation often used to predict bedload transport by turbulent flows. Moreover, many other

experimental studies of fans also reported flows that were not fully turbulent (e.g. Davies et al., 2003; Davies and Korup, 2007; Delorme et al., 2017, 2018; Van Dijk et al., 2012; Guerit et al., 2014; Hamilton et al., 2013; Reitz et al., 2010; Reitz and Jerolmack, 2012; Whipple et al., 1998). Although those models did not achieve Froude scaling, they successfully reproduced the fan-channel dynamics that are of interest to us.





## 2.3 Experimental runs

We varied the sediment supply periodicity between the four experiments. In Run 1, the sediment supply was constant at 5 g
s$^{-1}$. In Run 7, the supply rate was 10 g s$^{-1}$ for 5 minutes, followed by 0 g s$^{-1}$ for 5 minutes, and continued to oscillate between
these two extremes every 5 minutes, creating a 10-minute cycle of high- to zero-supply conditions. In Run 8, the cycle duration
doubled to 20 minutes (10 minutes each of high- and zero-supply). In Run 9, the duration doubled again to 40 minutes (20
minutes each of high- and zero-supply). These oscillations are summarised in Figure 2. Importantly, the mean sediment supply
rate in Runs 7-9 was 5 g s$^{-1}$, equalling the constant supply rate in Run 1. This meant that in all experiments, the same volume
of sediment and water was delivered in any 40 minute period, but with a different temporal distribution in the sediment supply.
The periods of the sediment supply oscillations in Runs 7-9 were chosen to explore the effects of sediment input duration, and
are not intended to scale to a specific event, season or climate oscillation in the field.

Each experiment was ~20 hours long, which was approximately the duration over which the fan prograded to the far walls
of the experimental table. For Run 1, we conducted three repeats in total; here we present data from repeat 2, which was closest
to the mean slope and area across the three repeats for that experiment. Runs 7-9 had only one repeat.

Using a length scale of 1:128, we approximated the experimental grain size distribution (GSD) from a surface gravel sample
collected in the channel at Three Sisters Creek fan, Canada, which is a typical gravel-cobble fan (located at 51.055108, -
115.333515; see also Figure A1). The experimental mixture ranged from 0.25 - 8 mm (Figure 3). This sandy GSD encouraged
subsurface flow, which sometimes generated seepage channels on the lower fan during the experiments. Such processes are
common on fans in the field; for instance, both down-fan channel narrowing and spring formation have been attributed to
infiltration on fans (Davidson et al., 2013; Kesel and Lowe, 1987; Woods et al., 2006).

All four experiments had a constant flow of 150 ml s$^{-1}$. This flow is approximately equal to the 20-year flood in the stream
where grain size data were collected. While the model is generic and does not represent a specific field prototype, the above
relation provides context for the size of this flow relative to the size of the sediment used in the experiments.

The sediment concentration was 3.6% by volume in the high-supply periods.The experimental grain size mixture was trun-
cated at 0.25 mm, omitting the finest ~40% of material in the field sample, so the true bedload sediment concentration could
be expected to be around 6% by volume. In mountain streams, bedload makes up between 10% and 99% of the total sediment
load, with a mean of 44.5% and standard deviation of 31.1% (see Table A1 for values reported in the literature). Consequently,
the maximum experimental truncated-bedload sediment concentration of 3.6% is roughly equal to a total volumetric sediment
concentration of 13.5% (7.9-45.0%) in the field.

The 150 ml s$^{-1}$ flow rate was relatively high, as was the sediment concentration. As a result, these experiments can be thought
of as analogues for a system that alternates between debris floods and clearwater floods; "clearwater" refers to flows with low
sediment concentration. This paper therefore explores the effect of abrupt, large-scale variations in the sediment concentration,
and the timing thereof.



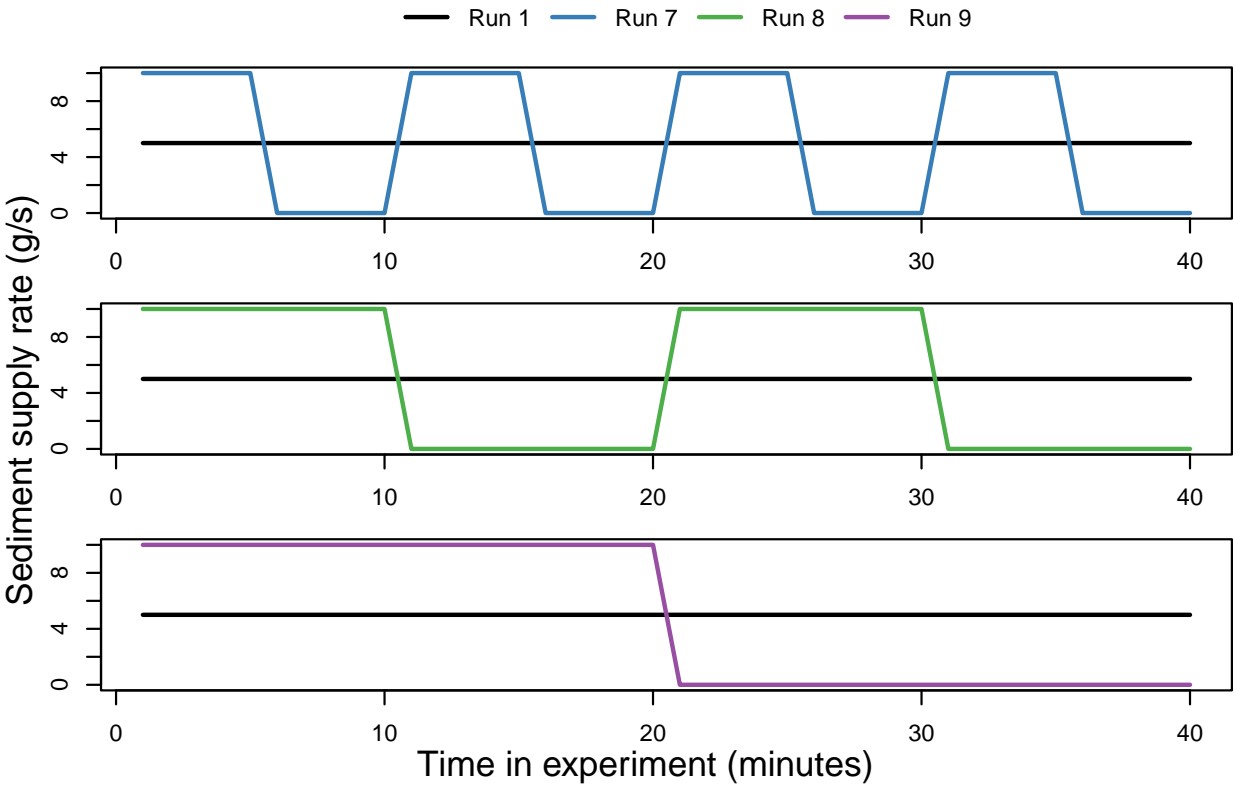

**Figure 2.** The durations of high-supply and zero-supply periods in Runs 1 and 7-9. The constant sediment supply rate in Run 1 equals the mean supply rate in all other runs.

## 2.4 Data analysis

Our photogrammetry system generated a topographic point cloud and co-registered 1 mm orthomosaic for each minute of each experiment. We interpolated the point cloud (using a nearest neighbour approach) to generate a 1 mm resolution digital elevation model (DEM). Following Leenman and Eaton (2021) and Leenman et al. (2022), only data from 12 hours and onward
were used, as some aspects of fan dynamics varied with fan size in the earlier stages of the experiment.

Fan slope was measured from 88 equally-spaced down-fan profiles, extracted from the DEM (see Figure A2 for their locations). For each profile, slope was taken from a linear regression of elevation against distance down-fan (profiles were quasi-linear). For each time-step, the representative fan slope was taken as the median of these 88 measurements.

Fan-head entrenchment was measured from arcuate cross-fan profiles extracted from each DEM at 0.25 m down-fan (see
Figure A2 for profile location). Fan-head entrenchment was measured as the difference between maximum and minimum elevations along the profile, for each time-step.





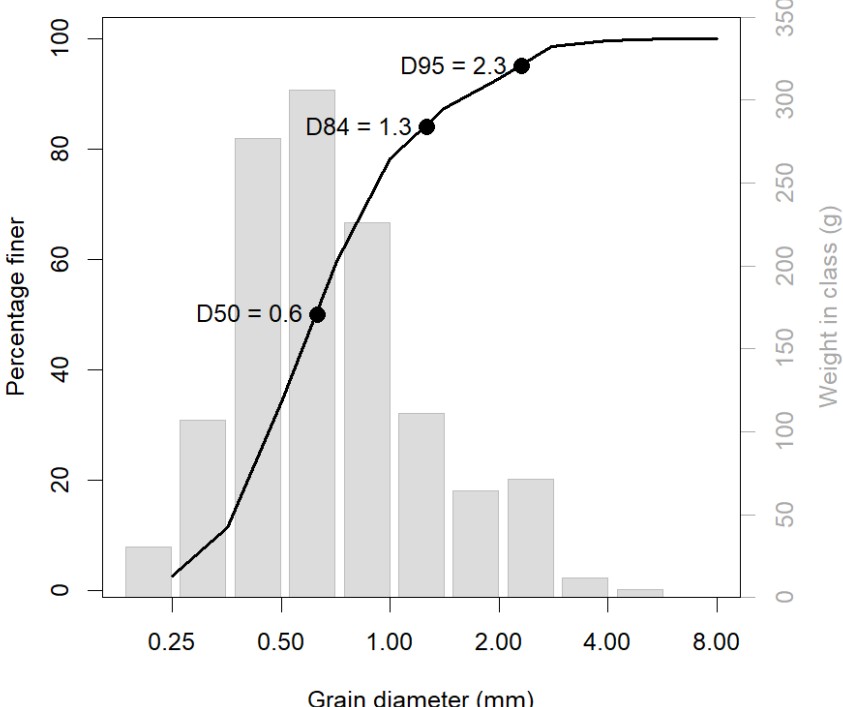

**Figure 3.** The grain size distribution (GSD) of our experimental sediment mixture. Cumulative distribution on primary $y$-axis; individual class weights on secondary $y$-axis.

The orthomosaics were processed to generate binary (wet-or-dry) maps of the fan and channels at each time-step. These flow maps quantified the proportion of the fan area covered by flow (referred to as the "wet fraction"). The flow maps were also used to count the number of channels at an arcuate cross-section 1 m down-fan (discounting seepage channels that were not connected to the fan-head by surface flow; see Figure A2 for cross-section location). In addition, change detection between successive flow maps revealed the area newly inundated in each minute, which was normalised by fan area to give $F_n$ (equation 1), the percentage of the fan newly inundated in a minute:

$$F_n\left(t\right) = \frac{Area\ newly\ inundated\ in\ previous\ minute}{Fan\ area\left(t\right)} \times 100 \tag{1}$$

Change detection between the DEMs produced DEMs of Difference (DoDs) that quantified the erosion and deposition in each minute. The DEMs were first smoothed with a $7 \times 7$ mm moving average filter (approximately the size of the largest sand grains). In the resulting DoDs, patches of erosion or deposition with a planform area of less than 2 cm$^2$ were discounted; elevation change of $< 2$ mm was also discounted. See Leenman (2021) for an uncertainty analysis of the DEMs, and further methodological detail.





## 3 Results

A general understanding of the fan responses to constant or oscillating sediment supply can be gained from the time-lapse videos (these show the fan from ~12 hrs onward): Run 1, Run 7, Run 8 and Run 9. Figure 4 also has links to these videos. Note that frames were collected at one-minute intervals for Run 1, and 10 s intervals for Runs 7-9, so the video speed differs. These videos demonstrate how the fan responded to constant sediment supply (Run 1) or sediment supply oscillations (Runs 7-9). Flow became more diverging when the sediment supply was turned on, and more channelised when the supply was cut

off. Rapid lateral migration and channel readjustment followed each change in the sediment supply.

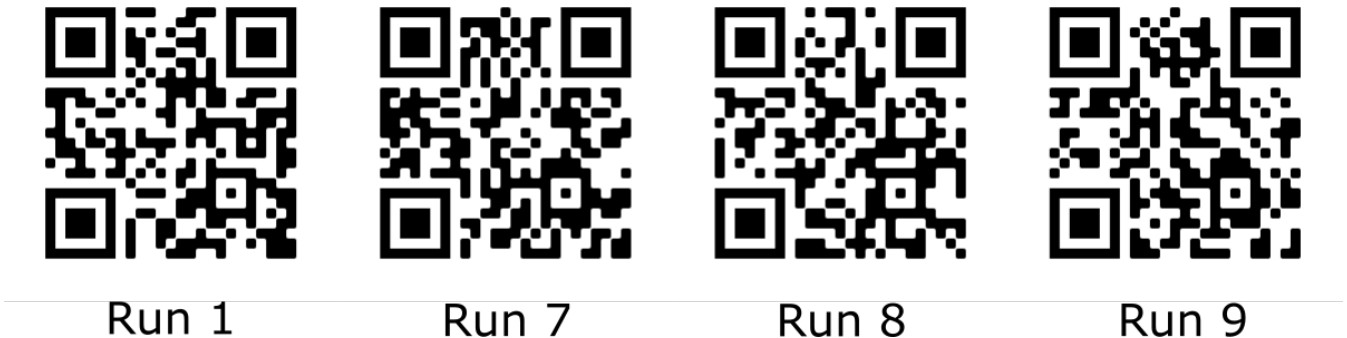

**Figure 4.** QR codes that link to the experimental time-lapse videos. The videos are also available at https://youtu.be/ML2LV28MQEM (Run 1), https://youtu.be/jXjWIkLU-7A (Run 7), https://youtu.be/T4JbZC9YkXQ (Run 8) and https://youtu.be/EcCWYGIbsqA (Run 9).

### 3.1 Fan morphology

Down-fan gradient is one of the simplest descriptors of fan morphology. As fans are self-formed, it provides a useful metric for their self-organised adjustment the input conditions or changes thereof. Figure 5 shows how the median down-fan slope differed across the four experiments, and how it adjusted during high- and zero-supply conditions.

Figure 5 shows that, compared to Run 1 (constant sediment supply), the fan was steeper with short-duration sediment supply oscillations (Run 7); it became flatter as the duration of the oscillations increased (Runs 8-9). The fan steepened during high-supply periods and regraded to a lower slope during zero-supply periods. This partly explains the trend in slope across the four experiments: during high-supply periods, sediment was deposited on the fan-head, steepening the fan, but during zero-supply periods the channel incised this material, lowering the fan slope. There was a lag of 2-3 minutes between the onset

of a new sediment supply rate and the geomorphic response. In the 10-minute cycle in Run 7, the 5-minutes of zero-supply was insufficient to completely incise the new material at the fan-head, leading to a fan that was steeper than it would be with constant sediment supply (Run 1). Conversely, in the 40-minute cycle in Run 9, the fan-head was deeply incised during the 20-minutes of zero-supply, reducing overall fan gradient. The spatial pattern of erosion and deposition, linked to these slope adjustments, is considered further in section 3.4.




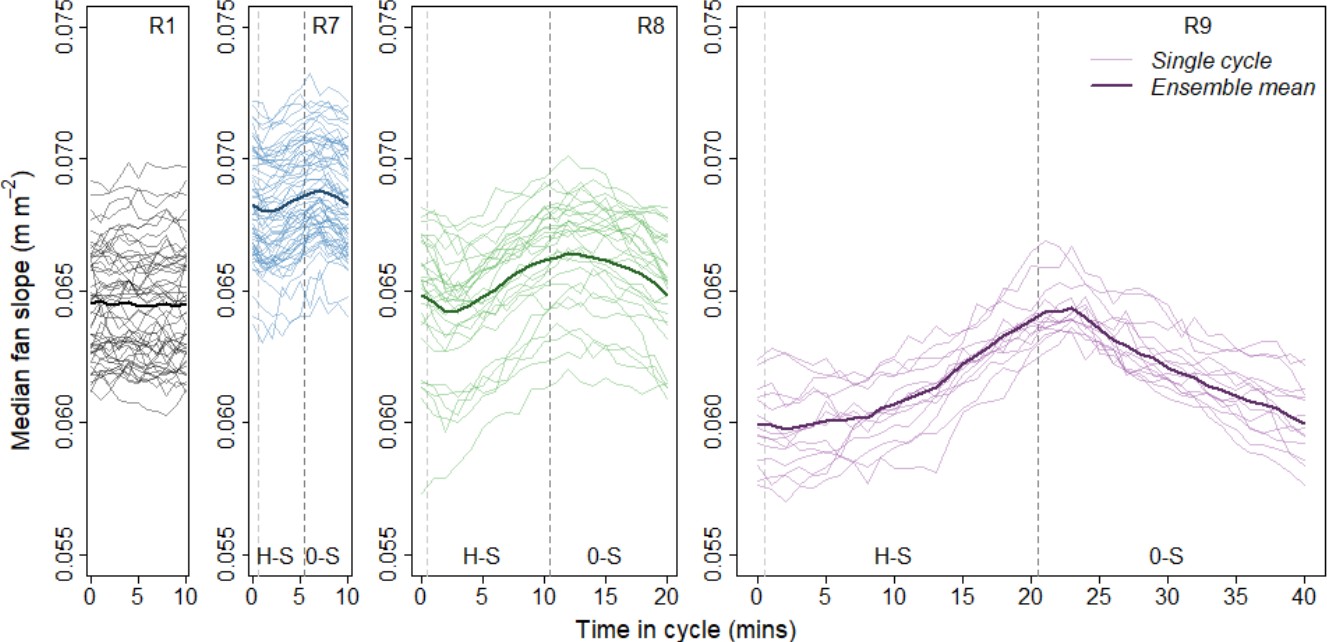

**Figure 5.** Median fan slope versus time in the sediment supply oscillation cycle. The thin lines represent each individual supply oscillation cycle; the thick line is the ensemble mean. H-S denotes the high-supply period; 0-S denotes zero-supply.

Figure 5 shows how median fan slope declined during zero-supply periods. However, the fan-head incision during zero-supply periods may in fact have meant that the slope along the fan-channel declined even farther than indicated by Figure 5. Table 1 suggests that the mean depth of fan-head trenching increased from Run 7-9 (see also Figures A5-A7). These values indicate that channel gradients (as opposed to fan gradients) at the end of the zero-supply periods were likely even lower than the median fan gradients shown in Figure 5.

**Table 1.** The mean (across all time-steps analysed) and maximum depth of fan-head trenching (in mm) at an arcuate cross-section across the fan-head, placed at 0.25 m down-fan.

| Run | Mean | Max. |
|-----|------|------|
| 7 | 9 | 16 |
| 8 | 11 | 19 |
| 9 | 14 | 26 |



## 3.2 Channel patterns

Channel patterns (and channel pattern change) provide a metric for how flow on the fan self-organises to transport the available sediment supply. Here, channel pattern is characterised using two variables: the number of connected channel threads at 1 m down-fan (Figure 6), and the portion of the fan occupied by flow (Figure 7).

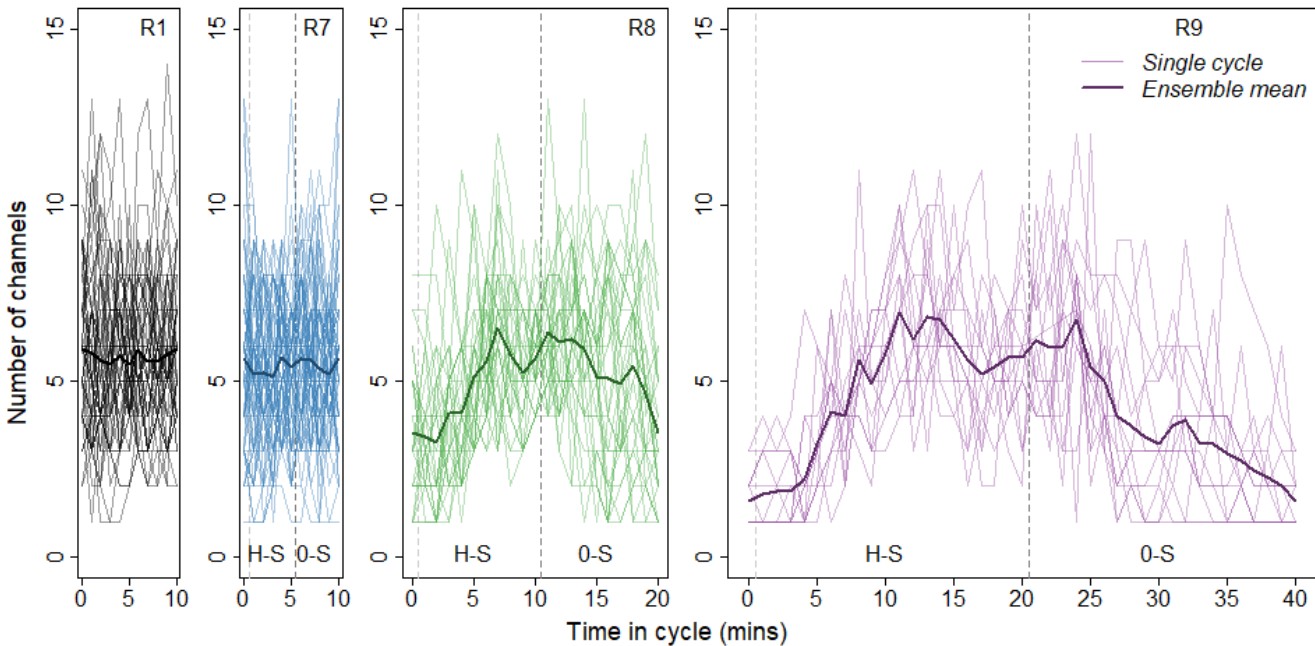

**Figure 6.** Number of **connected** channels versus time in the sediment supply oscillation cycle. The thin lines represent each individual supply oscillation cycle; the thick line is the ensemble mean. H-S denotes the high-supply period; 0-S denotes zero-supply.

Figure 6 shows that flow became more branching in high-supply periods, with the number of channel threads increasing. Conversely, flow became more channelised in zero-supply periods, with the number of channel threads decreasing.

The number of channel threads was slow to respond to a change in sediment supply rate. Consequently, the variability in the mean number of channels is similar between Run 1 (constant sediment supply) and Run 7 (10-minute supply oscillation cycle). Moreover, there is insufficient time for flow to channelise in the 5-minute zero-supply periods, so that flow maintains a divergent pattern in Run 7 with a fairly high number of channel threads (5-6 channel threads on average, at 1 m down-fan). In comparison, the 20-minute zero-supply duration in Run 9 means that fan-head trenching extends down-fan to the 1-m cross-section, bringing the mean number of channels down to around 2 at the end of the 40-minute cycle.

Comparing Figure 5 and Figure 6 indicates that fan gradient adjusted more readily than the channel pattern: fan gradient has a clear response to the changing sediment supply rate in Run 7, while the number of channels in Run 7 did not fluctuate more than in Run 1 (with constant sediment supply). This is likely because channels were counted at 1 m down-fan, but the channel





response to a change in sediment supply typically started at the fan-head and propagated down-fan to the 1 m cross-section (e.g. see time-lapse video for Run 9). Extracting the number of channels at the 0.5 m cross-section confirms that the fan-head channel pattern responded more readily to changes in the sediment supply rate (see Figure A3).

The second metric for channel pattern was the "wet fraction": the proportion of the total fan area that was inundated with flow in a given minute. Variations in the wet fraction (Figure 7) were more subtle than those in the number of channels, and provide a nuanced measure of how the flow pattern adjusted to changes in the sediment supply rate.

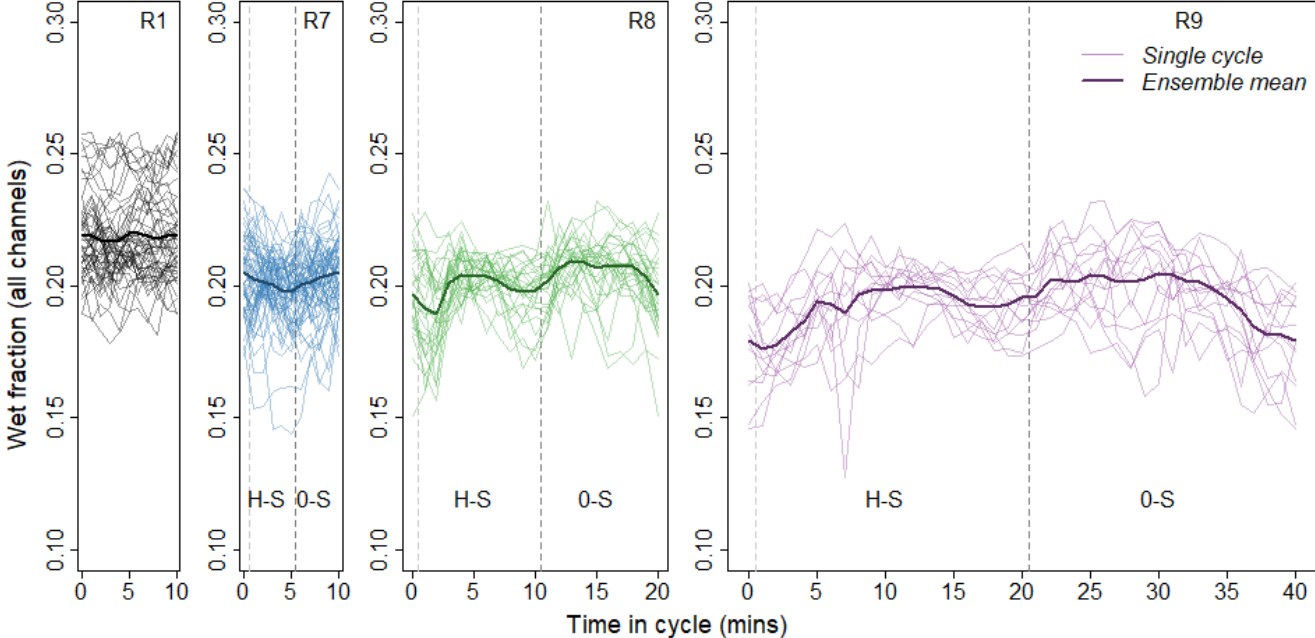

**Figure 7.** Wet fraction (measured from **all** channels), versus time in the sediment supply oscillation cycle. The thin lines represent each individual supply oscillation cycle; the thick line is the ensemble mean. H-S denotes the high-supply period; 0-S denotes zero-supply.

Figure 7 shows that, as with the number of channel threads, there was a lag of a few minutes before the wet fraction began to adjust to a change in the sediment supply. Consequently, the rapid supply oscillations in Run 7 (10 minute cycle) had little impact on the wet fraction; its variance was similar to that with constant sediment supply (Run 1). During Runs 8 and 9, the longer durations of high- and zero-supply show more clearly how the wet fraction adjusted to the sediment supply rate. At the start of high-supply periods, the wet-fraction increased after an initial lag, as flow widened and shallowed and/or slowed. A close examination of the experimental videos reveals that this increase was not through sheetflow or an abandonment of channelised flow, but through flow divergence into numerous channel threads; these were often poorly defined and interconnected, as with braided streams. Toward the end of the high-supply period, the wet fraction decreased, suggesting that flow ultimately organised into fewer, deeper channels (although still more channels than during zero-supply periods). Close examination of Figure 6 supports this notion.



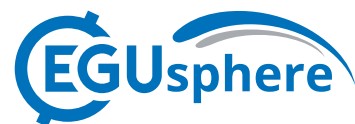

At the start of zero-supply periods, the wet fraction increased briefly, showing that a larger portion of the fan was inundated as flow re-adjusted to the reduced sediment concentration. However, the number of channel threads decreased during this period, as did the sector of the fan occupied by the flow (Figure 6 and Figure A4). Their decrease implies that flow was collecting into fewer channels, but that those channels were initially shallow (or slow) and wide as the total area of flow was high. Toward
the end of the zero-supply period, the wet fraction decreased, suggesting that channels were becoming deeper (or faster) and narrower as the flow area decreased. These changes can be seen in the experimental videos as well.

### 3.3 Lateral mobility

In addition to characterising channel patterns, we also monitored channel *change*. Here, we characterise lateral channel mobility by measuring $F_n$, the percentage of the fan area that is newly inundated in a given minute. High $F_n$ values imply either avulsion,
a rapid channel sweep across the fan, or a rapid change in channel pattern (e.g. divergence from single- to multi-threaded flow). Conversely, a low $F_n$ reflects a relatively stable channel.

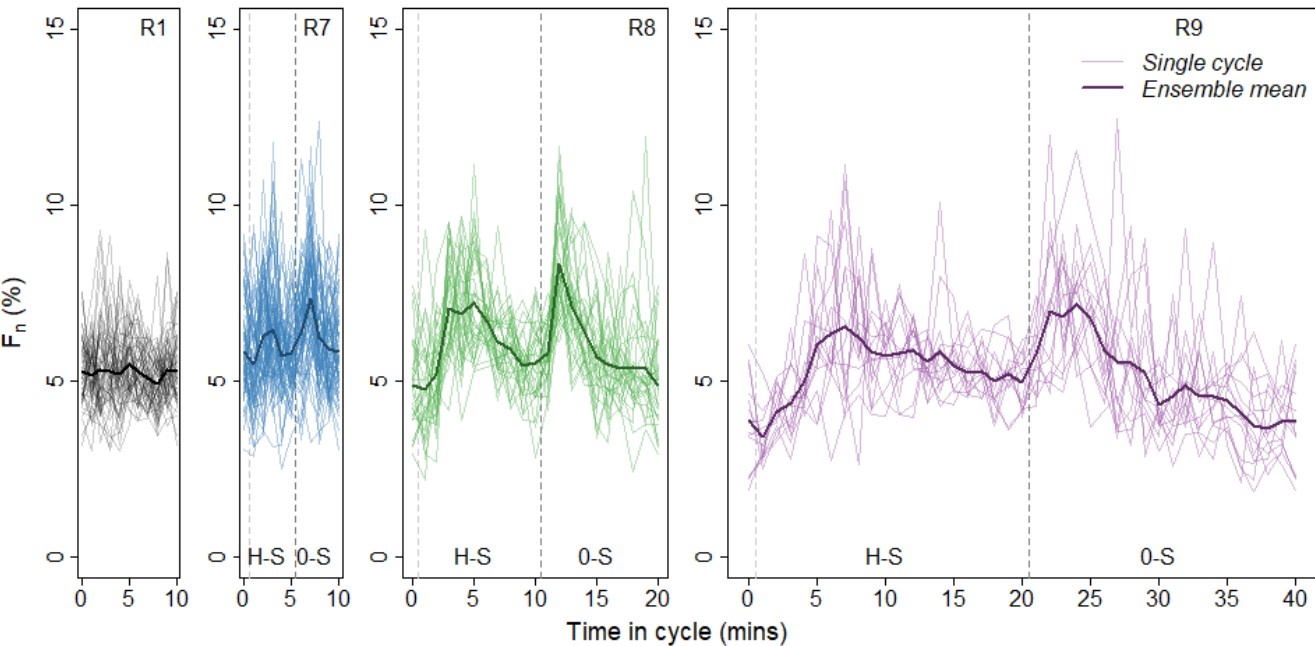

**Figure 8.** $F_n$ versus time in the sediment supply oscillation cycle. The thin lines represent each individual supply oscillation cycle; the thick line is the ensemble mean. H-S denotes the high-supply period; 0-S denotes zero-supply.

Figure 8 shows how lateral mobility ($F_n$) varied during each sediment supply oscillation cycle. After any change in the sediment supply, lateral mobility peaked, with the channel adjusting rapidly to the altered sediment concentration. This peak decayed gradually during both high- and zero-supply periods. Lowest $F_n$ values, implying the most stable channel pattern,
were attained in Runs 8 and 9 at the end of the zero-supply periods. This stability reflects an incised channel, which must be





filled with sediment in the next high-supply period before rapid migration can re-commence. Such channel filling thus delayed the onset of peak mobility in the following high-supply period, with the high-supply mobility peak occurring later in Runs 8 and 9, as those runs had longer zero-supply periods in which the channel became more entrenched at the fan-head. These results mirror a set of experiments by Vincent et al. (2022), in which longer duration low-flow periods between debris floods

increased the time required for a debris flood to cause avulsion.

### 3.4 Morphologic reworking

In addition to the lateral mobility computed by comparing flow maps, we calculated rates of vertical change (i.e. erosion and deposition) by performing change detection between successive DEMs. Figure 9 shows the volumes of sediment deposited or eroded in each minute of the sediment supply oscillation cycle.

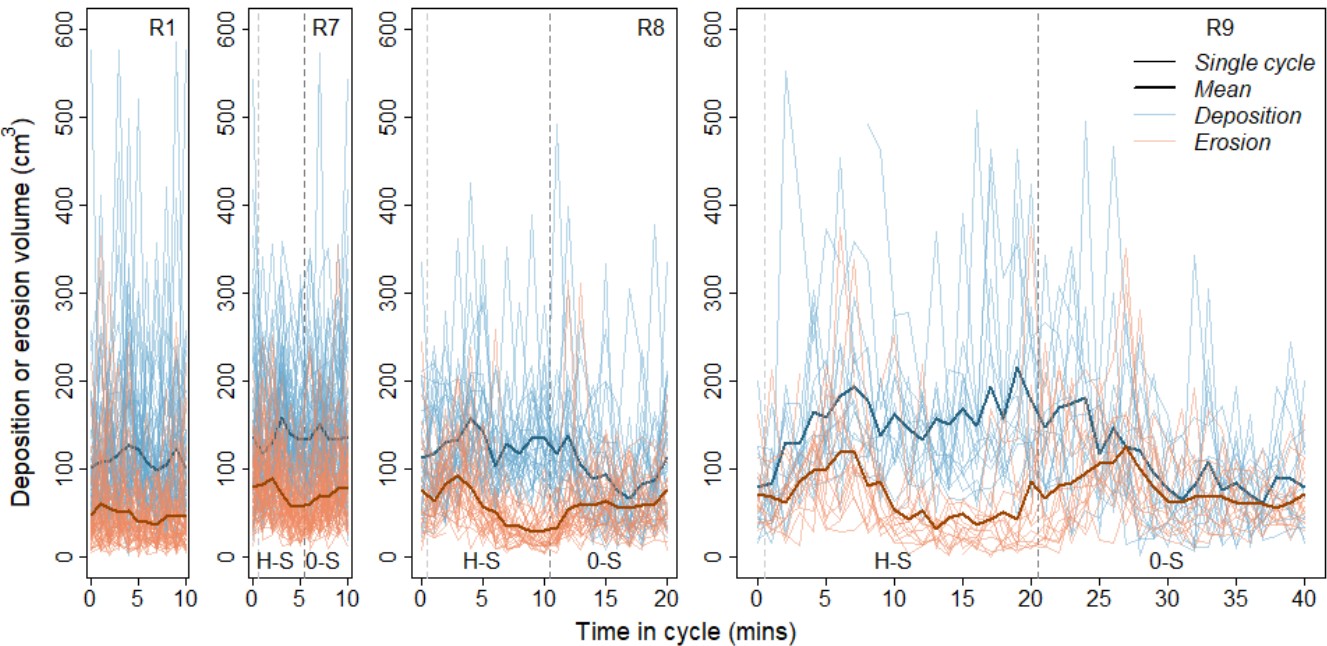

**Figure 9.** Deposition (blue) and erosion (red) volumes versus time in the sediment supply oscillation cycle. The thin lines represent each individual supply oscillation cycle; the thick lines are the ensemble means. H-S denotes the high-supply period; 0-S denotes zero-supply

.

Figure 9 indicates that the addition of short sediment supply oscillations in Run 7 increased the erosion and deposition rates on the fan, compared to Run 1 with constant sediment supply. The variance in erosion and deposition was similar in Run 7 and Run 1, although the timing of peaks in Run 7 followed the same pattern as Runs 8 and 9, described below.

In Runs 8-9, deposition increased during high-supply periods. The increase was gradual, with peak deposition coming several minutes after the onset of high-supply. This reflects the fact that deposition began in the feeder channel upstream of the fan-



head, which buffered the fan from the immediate effect of changes in environmental conditions in the same way that a confined upstream reach would in a natural system. When the sediment supply was turned off, deposition rates remained high for a few minutes as sediment in the feeder channel was mined. Deposition rates then decayed toward the end of the zero-supply period; this decline was most evident in Runs 8 and 9.

Erosion also increased at the start of the high-supply period, even though deposition rates were high. This reflects a period
of channel adjustment in response to the sudden increase in sediment supply. After the initial peak, erosion decreased to a minimum in the second half of the high-supply period. When the zero-supply period commenced, erosion accelerated, reflecting the rapid lateral migration (Figure 8) at this time. After the initial increase, erosion rates stabilised during the zero-supply period. This was particularly evident in Runs 8 and 9, where erosion and deposition rates were approximately equal toward the end of the zero-supply. Their similarity implies that, through fan-head trenching during the zero-supply period, the
fan reached a form of equilibrium with the imposed flow rate and the supply of sediment from incision. Although erosion and deposition occurred, there were no major peaks, suggesting that there was little channel reorganisation. Instead, a condition of relative stability persisted, with the channel gradually incising the fan-head and depositing sediment on the lower fan.

The average spatial patterns of erosion and deposition are revealed in Figure 10, which demonstrates the effects of length-ening the sediment supply oscillations. During high-supply periods (top row), deposition was concentrated at the fan-head,
and particularly in the fan-head trench (most visible in Runs 8 and 9). Downstream of this deposition was a focused erosion zone. As the duration of the sediment supply oscillation cycle increased, this deposition-erosion couplet extended farther downstream. The deposition zone became more elongate, emphasising the filling of the fan-head trench, while the erosion zone shifted farther down-fan.

During zero-supply periods, trenching at the fan-head was again clear, with a zone of focused erosion at the fan-head. The
erosion zone and fan-head trench extended down-fan as the duration of the zero-supply period increased. A zone of focused deposition radiated from the downstream end of the fan-head trench, and is particularly visible in Runs 8 and 9. This pattern indicates that, during the longer zero-supply periods in Runs 8 and 9, the fan-head trench incised and acted as a conduit for sediment eroded from the feeder channel. The morphology of the fan-head trench is also visible in the cross-fan topographic profiles in Figure A5 - A7.

The cumulative effect of this coupled fan-head trenching and lower-fan deposition can be viewed by comparing the fan shapes in Figure 10. Each DoD compilation in this figure was masked to the fan planform at 18 hours of experimental run time; the same total volume of water and sediment had been delivered to each fan. Nevertheless, as the durations of sediment supply oscillations increased, the fan shape became more elongate: the mid-fan (axial) radius increased from Run 7 to 9, while the side radius remained comparable across all runs. This reflects fan-head trenching and sediment transfer from the upper to
lower fan during the zero-supply periods, which lengthened the central fan.





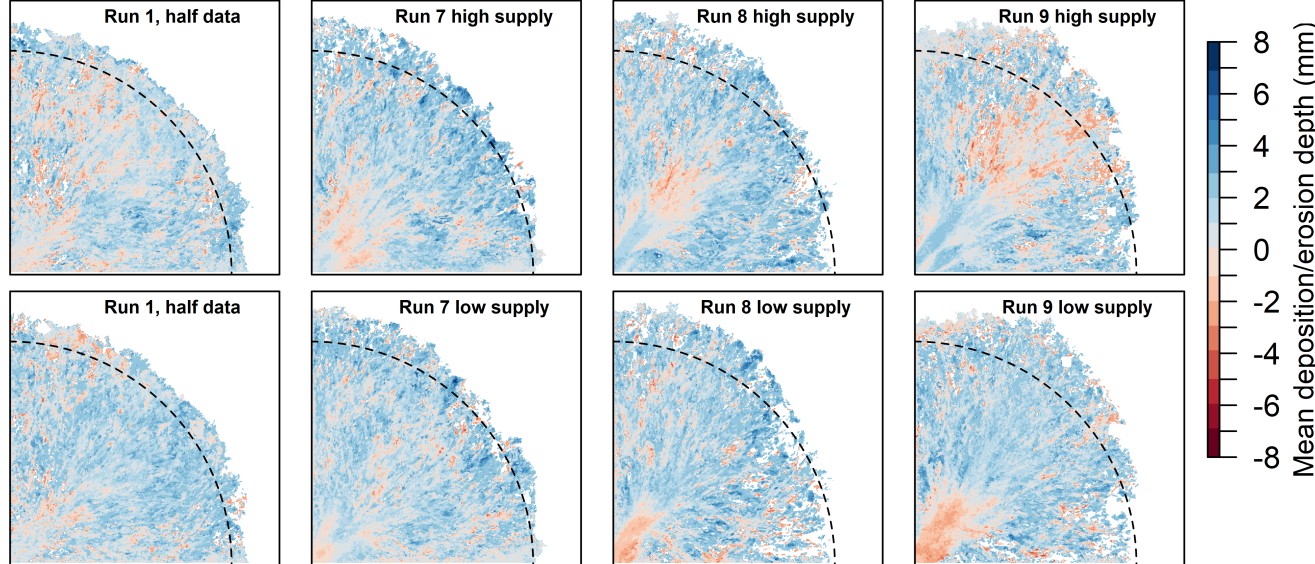

**Figure 10.** Spatial patterns of erosion (red) and deposition (blue). All one-minute DoDs for high-supply (above) and zero-supply periods (below) were averaged to show the typical pattern. Data from Run 1 are split into the same groups as Run 7, although the former had no sediment supply oscillations; the Run 1 data show how deposition and erosion varied randomly in comparison to the spatially organised erosion and deposition in Runs 7-9. The semicircle at 2 m down-fan shows how the central fan became more elongate as the sediment supply oscillations lengthened.

## 4   Discussion

By alternating between a high sediment supply rate and no sediment supply, and by varying the duration of oscillations, these experiments permit the investigation of two key phenomena on alluvial fans. First, that of how fan-channels respond to abrupt changes in sediment concentration. Second, that of how the frequency and duration of sediment supply oscillations affect fan dynamics and morphology. Through comparing constant- and oscillating-supply experiments, this paper also considers the (im)propriety of substituting a constant sediment supply for a naturally variable one when modelling alluvial fans in the lab.

These experiments have demonstrated how fan-channels respond to an abrupt increase in the sediment supply rate (and therefore, the sediment concentration). The primary responses were steepening, an increased number of channels, and increased inundated area. This was accompanied by a rapid increase in lateral mobility that tailed off as the flow pattern stabilised, and by a peak in erosion rate during this channel adjustment phase. Deposition was generally high during high-supply periods.

When sediment supply was abruptly cut off, the fan responded through slope reduction and channelisation. The inundated area was high at first, while flow adjusted toward a single channel, but then decreased as that channel incised. Similarly, there was a steep but short-lived peak in lateral mobility as the flow pattern adjusted. This pulse of lateral mobility was accompanied



by an increase in erosion, that stayed high during the zero-supply period due to fan-head incision. Meanwhile, the deposition
rate gradually decreased, eventually equilibrating with the erosion rate in the runs with longer zero-supply periods.

Extending the length of the sediment supply oscillations altered fan morphology, even though the mean sediment supply
rate was unchanged. Short oscillations led to sediment accumulation at the fan-head, steepening the fan in comparison to the
constant supply experiment, and generating diverging, laterally active channels. Conversely, longer oscillations allowed the
channel to incise the fan-head, shifting the zone of geomorphic activity farther down-fan and giving rise to a flatter gradient
and more elongate fan shape.

## 4.1    Sediment concentration variability on fans in the field

These "similarity-of-process" experiments are a simple but useful model of fan dynamics (Hooke, 1968a; Paola et al., 2009).
The different scenarios explore the limits of fan behaviour: while there are few situations in which fans abruptly alternate
between high-supply and clearwater floods, wide temporal variations in sediment concentration do occur on fans and in steep,
gravel-bed streams (e.g. Garcia et al., 2000; Habersack et al., 2001; Hayward and Sutherland, 1974; Kuhnle, 1992; Mao
et al., 2014; Meyer and Wells, 1997; Reid et al., 1985; Wasson, 1974; Wells and Harvey, 1987). Our experiments show how
fan channels respond to some of the widest possible variations in sediment concentration; one could expect lower-amplitude
variability to generate dampened but qualitatively similar responses.

On fans in the field, wide variations in sediment concentration can occur during a single flood event, or between successive
events. For instance, sediment exhaustion over the course of a storm can rapidly decrease sediment concentration during a flood
event (Meyer and Wells, 1997; Wasson, 1974; Wells and Harvey, 1987). Similarly, debris flow control nets are designed to trap
coarse sediment but transmit water and fine sediment (Wendeler et al., 2007; Wendeler and Volkwein, 2015); therefore, their
emplacement in source basins can lower the bedload sediment concentration in flows reaching fans. Lower amplitude sediment
concentration variations have also been observed during floods in gravel bed rivers (e.g. Garcia et al., 2000; Habersack et al.,
2001; Hayward and Sutherland, 1974; Kuhnle, 1992; Mao et al., 2014; Reid et al., 1985).

The experiments presented in this paper provide a controlled analogue to the scenarios described above; the experimental
results indicate how fans in the field may respond to these different scales of variability. For instance, the changes in channel
pattern and lateral migration in Run 7 (see Figures 6 and 8) were minor, suggesting that short-term fluctuations in the sed-
iment concentration do not strongly influence channel pattern. Although Run 7 had the steepest slope (Figure 5), this likely
reflects the high-amplitude of the sediment supply rate variations, from 0 to 10 g s$^{-1}$. A preliminary experiment with lower
amplitude variations but the same mean sediment supply rate and oscillation duration generated a less-steep fan (Figure A8).
Consequently, it seems that small-scale, short variations in sediment concentration during a flood will have little influence on
the overall channel pattern or fan gradient.

Conversely, the experiments show that longer-duration or larger-amplitude variations in the sediment concentration do im-
pact fan-channel adjustment during high flows, and may have longer term implications for fan morphology. For instance,
depletion of the sediment supply to a fan during a storm event may mean the channel responds through rapid lateral migration



and erosion, as in Runs 8 and 9 (Figures 8). Similarly, emplacement of a debris retention structure which reduces sediment concentration during subsequent flood events might lead at first to lateral migration, and eventually to fan-head trenching.

While these experiments highlight the impact of rapid changes in sediment concentration, they also underscore the impor-
tance of clearwater flow for shaping fan and channel morphology, and particularly the fan-head trench. Such trenches are
ubiquitous in natural fans of varying ages in varying climates (e.g. Bowman, 1978; Bluck, 1964; Davies and Korup, 2007;
Dorn et al., 1987; Harvey, 1987; Mather and Hartley, 2005, among others). Previous experimental studies have suggested that
an incised fan-head trench can develop in the absence of external perturbations; that is, when water and sediment are supplied
at constant rates (Clarke et al., 2010; Van Dijk et al., 2012; Schumm et al., 1987; Whipple et al., 1998; Zarn and Davies, 1994).
Nevertheless, in the constant flow and sediment supply experiment presented here (Run 1), a single-channel never persisted for
long enough to allow significant fan-head trenching.

In contrast, a distinct fan-head trench did develop in Runs 8-9, with longer sediment supply oscillations. It is interesting to
note that there was no notable fan-head trench in a related set of experiments with short-duration flow variability (published
in Leenman et al., 2022) or in the experiment with short-duration sediment supply variability (Run 7). These results suggest
that longer-term sediment concentration variations (and particularly periods of lower-than-average sediment supply) encourage
the formation of an incised fan-head trench. Figure 6 suggests one explanation for this phenomenon: the number of channels
decreases during periods of low sediment concentration, as flow collects into a few main channel threads. This concentrates
the erosive activity of the channel into a narrow sector of the fan (see also Figure A4), which likely enhances the rate of
down-cutting and trench formation.

In our experiments and in some active fans in the field, the fan-head trench fills when sediment supply is high; in nature, this
may occur when landslides temporarily raise the sediment supply (e.g. Davies and Korup, 2007). A fan-head trench does not
necessarily require a clearwater flood event in order to develop: experiments by Vincent et al. (2022) showed that even low flow
periods with no sediment supply were capable of incising a fan-head trench, given a sufficiently long period. An additional
implication of the experiments in this paper, supported by the work of Davies and Korup (2007) and Vincent et al. (2022), is
that the impact of floods with high sediment concentration is mediated by the depth and length of the fan-head trench, as this
depression must be filled before avulsion can occur at the fan-head.

In the field, depositional events on fans have formed deposits of a comparable size to that generated by the high-supply
periods in these experiments. In Runs 7-9, fan-head trenches were on average 9-14 mm deep respectively; over all areas of
topographic change (including erosion) the mean depth of topographic change in high-supply periods was 2-4 mm respectively.
The ratio of deposition depth to trench depth therefore ranged from 0.2-0.3. On fans in southwest New Zealand, Davies and
Korup (2007) observed depositional lobes of ~1-5 m thick on fans with trenches ~10 m deep, giving similar ratios of 0.1-0.5.
Following the 2010 landslide and debris-flow at Mount Meager, British Columbia, debris-flow sediment completely filled a
creek in an incised trench of 5-15 m depth (a ratio of 1.0), thinning downstream to around 2 m (a ratio of 0.1-0.4) (Guthrie
et al., 2012). In northeast New Zealand, post-logging deposition on fans ranged from <1 to ~10 m in depth, in one case
filling an incised fan-head trench by 12 m (Leenman and Tunnicliffe, 2020). These comparisons demonstrate how the size
of depositional events relative to fan channel dimensions are similar in the model and some field examples. Moreover, the





comparisons highlight the relatively large depth of fan-head incision that can occur during prolonged periods of clearwater flow.

Parallels can be drawn between our experimental fans and natural fans formed by basins that are supply- or transport-limited. For instance, a transport-limited basin is less likely to have frequent or prolonged clearwater floods. Such a basin is most similar to Run 1 or 7, and one might therefore expect it to generate a steeper fan with diverging, laterally active channels, all other things being equal. Conversely, a supply- or "weathering-limited" basin (Bovis and Jakob, 1999) might have more frequent or prolonged clearwater floods, or floods with low sediment concentration. It is therefore more similar to Runs 8 or 9, and one might expect it to generate a fan with a lower gradient and more incised fan-head trench that is relatively stable, punctuated by periods of high lateral activity and more divergent flow when hillslope erosion generates peaks in the sediment concentration (such as that observed on the Poerua fan by Davies and Korup (2007)).

Climate change is increasing the frequency and severity of extreme weather events that lead to floods (IPCC, 2022). In transport-limited basins, such changes could increase the frequency of flood events with high sediment concentration, making fan behaviour more similar to Run 1 or 7. Conversely, in supply-limited basins, this hydroclimatic change could increase the frequency of floods capable of reworking the fan, and may have less impact on the sediment supply (depending on the nature of hillslope erosion). Such a change might make fans more similar to Runs 8 or 9.

## 4.2 Sediment supply oscillations and fan form

Studies of alluvial fans have sought for many decades to relate fan area and slope to source basin morphometry (e.g. Al-Farraj and Harvey, 2005; Beaumont, 1972; Bull, 1964; Crosta and Frattini, 2004; Denny, 1965; De Scally and Owens, 2004; Harvey, 1984; Hooke, 1968b; Kostaschuk et al., 1986; Lecce, 1991; Melton, 1965; Milana and Ruzycki, 1999; Oguchi and Ohmori, 1994; Saito and Oguchi, 2005; Silva et al., 1992; Stokes and Mather, 2015; Stokes and Gomes, 2020; Tomczyk, 2021). A key goal of those studies was to untangle the sources of scatter in such relationships, thereby elucidating the controls on fan slope and area. For instance, scatter has been attributed to fan setting (e.g. tributary-junction vs mountain-front; Al-Farraj and Harvey, 2005), down-stream valley width (Stokes and Mather, 2015), and depositional process (debris flow vs fluvial; De Scally and Owens, 2004).

While studies of fan and catchment morphometry have contributed a great deal to our understanding of alluvial fans, linking catchment and fan variables rests upon the assumption that fans have reached (or will reach) an equilibrium with their source catchment. This is not always the case; the sediment supply to fans oscillates at periods ranging from single events (e.g. Cabre et al., 2020) to orbital cycles (e.g. Blechschmidt et al., 2009). In fact, this temporal variability gives rise to the concept of alluvial fans as environmental "indicators" that record changes in sediment supply (Harvey, 2012). Our experiments show how, over many successive cycles, such oscillations in the "upstream" conditions of an alluvial fan can alter fan morphology, even for the same average sediment supply rate. Consequently, our results reveal another potential source of scatter in relationships between basin and fan morphometry: the periodicity of sediment inputs.

The flattening and lengthening of our experimental fans with increasing oscillation duration (see Figure 10 in particular) also suggests that, in the field, we can infer something of the sediment-supply histories of fans based on their elongation. For





instance, fans that have a shorter, more "stacked" morphology might form from high-frequency oscillations. Conversely, fans with more "telescoping" morphologies (like that of Run 9) might reflect a more intermittent sediment supply, with longer-term periods of high and then low sediment supply. In particular, longer periods of low sediment supply likely allow for sediment redistribution from the upper to lower fan, forming a fan-head trench and "telescoping" lower fan. Figure 10 suggests that

the fan-head trench becomes more elongate (and the intersection point farther down-fan) as the duration of a low sediment supply period lengthens. One example of a set of natural fans with elongate, "telescoping" morphologies is the mountain front fans along the west of the Musandam Mountains (Al-Farraj and Harvey, 2005). Prolonged periods of higher monsoonal rainfall throughout the Late Quaternary have driven long-term changes in the sediment supply to these fans (Blechschmidt et al., 2009), which may account for their telescoping morphology.

### 4.3   A representative sediment supply rate?

These experiments show how sediment supply fluctuations can affect fan morphology and channel patterns. However, they also highlight a problem in the common approach to physical models of alluvial fans: that of constant sediment supply as an approximation for a range of variable sediment supply rates in the field. While the temporal variations imposed in Runs 7-9 were almost as simple as possible, they nevertheless generated fans with morphology, channel patterns and behaviour that were

different from Run 1, with constant sediment supply. Moreover, fan morphology varied systematically with the duration of high- and zero-supply periods. These findings indicate that constant sediment supply rates poorly represent fans in the field that are governed by episodic sediment supply.

It might be possible to find a "representative" sediment supply rate which generates the same fan slope as would a series of episodic sediment delivery events. For instance, median fan slope was similar in Run 1 (constant sediment supply) and Run 8

(20-minute oscillation cycle). One could infer that the 20-minute cycle in Run 8 may represent some kind of "characteristic event periodicity" for the given sediment supply and flow rates, with the 10-minute cycle being too short (and therefore producing steeper fans) and the 40-minute cycle being too long. However, while Runs 1 and 8 produced similar fan slope, geomorphic activity was much more variable in Run 8, with higher extreme values of lateral and topographic change. Consequently, when seeking to model natural hazards on fans, it might be appropriate to choose a higher "representative" sediment supply, or to

include oscillations about the mean. Although doing so may mis-represent fan slopes, it is more likely to capture the extreme values of erosion or lateral migration which are of interest for understanding and managing natural hazards.

Other experimental studies have explored how sediment supply alters fan gradients and channel dynamics (Ashworth et al., 2004; Bryant et al., 1995; Delorme et al., 2018; Whipple et al., 1998). However, those experiments featured constant sediment supply rates. The experiments presented here show that both fan gradient and channel dynamics vary systematically with the

duration of sediment supply oscillations. These results raise the possibility of equifinality: natural fans with steeper gradients and laterally active channels may result from high sediment supply, from rapid fluctuations between high and low sediment concentrations, or from some combination of both these influences. Moreover, it is possible that long-term average sediment supply and the intermittency of sediment inputs could covary in natural fans, making these two controls difficult to separate. Series of shorter and longer sediment supply likely occur as well, which may generate legacy effects depending on their



sequence. Further experiments varying both the mean sediment supply and the duration of supply oscillations could aid in
addressing these questions.

## 5    Conclusions

This paper presents the results of four experiments with the same mean sediment supply rate and a constant flow. Three of
the experiments alternated between a high supply rate and no sediment supply, with the duration of the oscillations varying
between experiments. The experiments addressed a) how fan-channels respond to abrupt changes in sediment concentration,
and b) how fan dynamics and morphology vary with the duration of sediment supply oscillations.

The experiments showed that, when sediment concentration increased abruptly, fans steepened and flow diverged into more
channel threads, inundating a larger fraction of the fan. Lateral mobility and erosion rates were high at first, before the channel
pattern stabilised. Deposition rates remained high while sediment concentration was high. Conversely, when sediment con-
centration decreased abruptly, fans adjusted through slope reduction and channelisation. The inundated area, lateral mobility
and erosion rate were high at first, until flow adjusted toward a single-channel state. The erosion rate remained elevated while
sediment concentration was low, due to fan-head trenching.

The duration of high- and zero-supply periods systematically affected fan morphology and channel dynamics. Short-term
oscillations promoted fan-head deposition, steepening the fan and generating diverging, laterally active channels. Long-term
oscillations promoted fan-head incision, shifting the zone of geomorphic "activity" down-fan and generating a flatter, more
elongate fan.

These experiments, conducted with a constant "flood" flow, highlight how sediment concentration governs the geomorphic
impact of a flood event on an alluvial fan. Debris retention structures designed to pass floodwater only, or natural sediment
exhaustion during floods, prompt a reduction in sediment concentration. These experiments indicate the possible channel
response in such a scenario.

The varying durations of high- and zero-supply periods in the experiments also demonstrate how different basin conditions
might generate different fan morphology. Transport-limited basins, more similar to the short-oscillation experiment presented
here, might be expected to have steeper slopes and more diverging flow. Supply-limited basins, more similar to the long-
oscillation experiments here, might have gentler slopes and more channelised flow. The exact responses are likely to depend on
the sediment concentration (and its variability) in the flow to fans, which was monitored only indirectly in this study by using
sediment supply as a proxy.

Finally, the duration of sediment supply oscillations produced systematic variation in fan slope and area, even though all
experiments had the same mean sediment supply rate. This raises the question of how closely experimental fans built with
constant sediment supply can be said to represent fans in the field. Future studies of alluvial fans could consider supply
variability when modelling how sediment supply affects alluvial fans.



*Code and data availability.* Basic data processing steps were conducted using the code at https://github.com/a-leenman/phd_code. Code to conduct further analysis and produce the figures can be found at https://github.com/a-leenman/Leenman_Eaton_2022. Data underlying the figures can be downloaded at https://doi.org/10.5281/zenodo.6499441. The raw data from these experiments is still under analysis for subsequent publications; please email the corresponding author if you would like a copy.

*Video supplement.* Time-lapse videos of the experiments are available at https://youtu.be/ML2LV28MQEM (Run 1), https://youtu.be/jXjWIkLU-7A (Run 7), https://youtu.be/T4JbZC9YkXQ (Run 8) and https://youtu.be/EcCWYGIbsqA (Run 9).



**Appendix A: Supplementary Material**

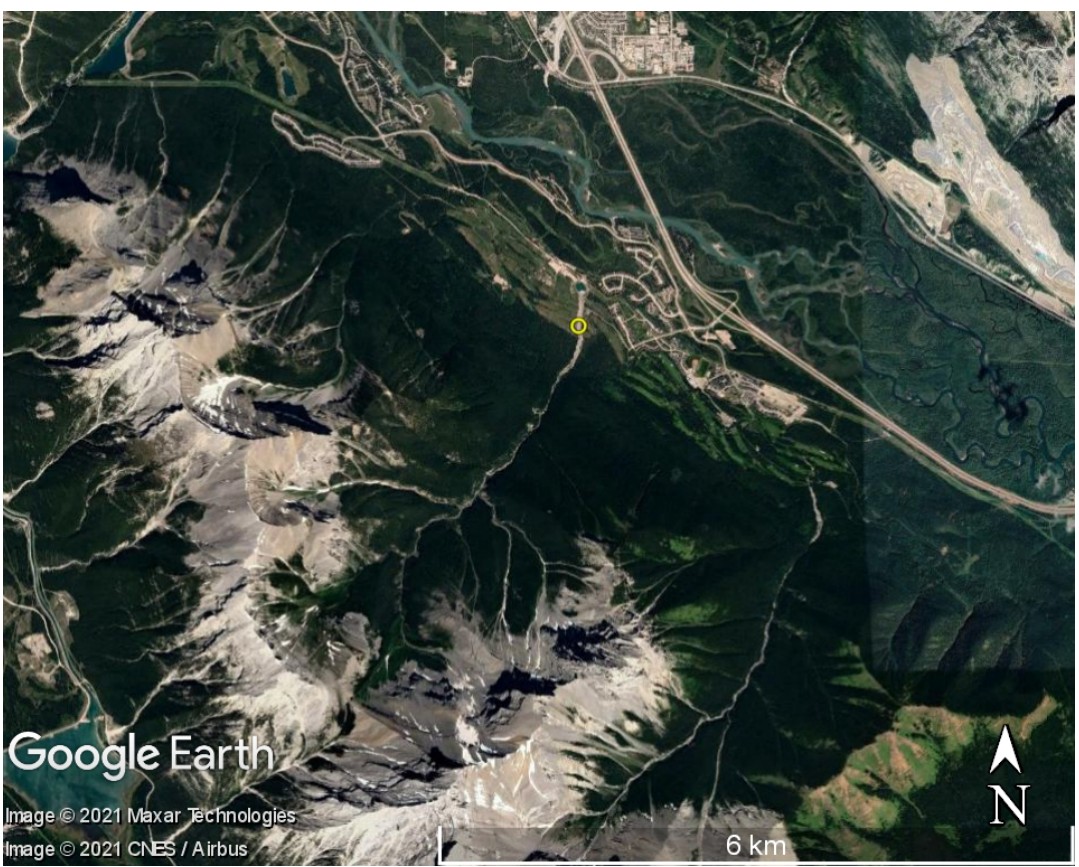

**Figure A1.** Three-Sisters Creek catchment and fan, on the southwest side (right bank) of the Bow River, Alberta, Canada, near the town of Canmore. The surface grain size sample was collected at approximately the location of the yellow circle. Photo © Google Earth.



**Figure A2.** Locations of down-fan and cross-fan transects where measurements were taken. The 88 down-fan transects (black lines) were used to measure fan gradient. The arcuate cross-fan profile at 0.25 m down-fan (inner white line) shows where fan-head entrenchment measurements were conducted. The cross-fan profile at 1 m down-fan (outwer white line) shows where the number of channel threads was measured. Underlying hillshade (displayed with 3 × elevation exaggeration) is from Run 9 at 19 hrs of experimental time, close to the end of the experiment.





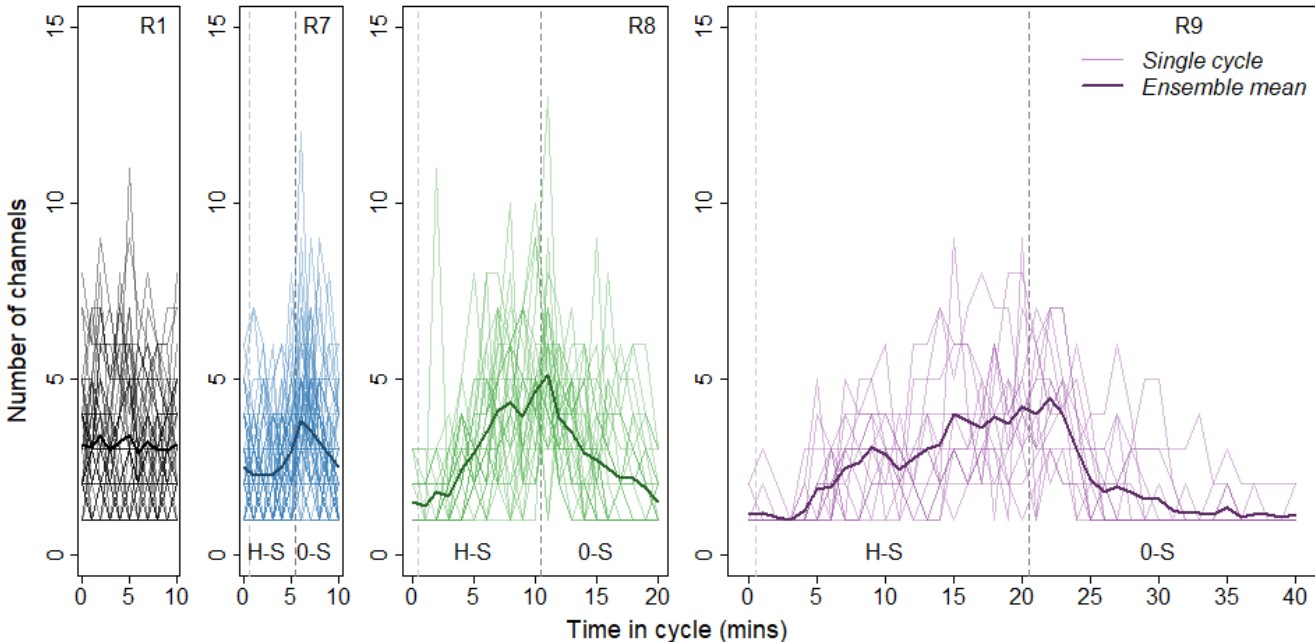

**Figure A3.** Number of **connected** channels at 0.5 m down-fan, versus time in experimental cycle. Thin lines represent each individual high-zero sediment supply cycle; thick line is the ensemble mean. H-S denotes high-supply period; 0-S denotes zero-supply.





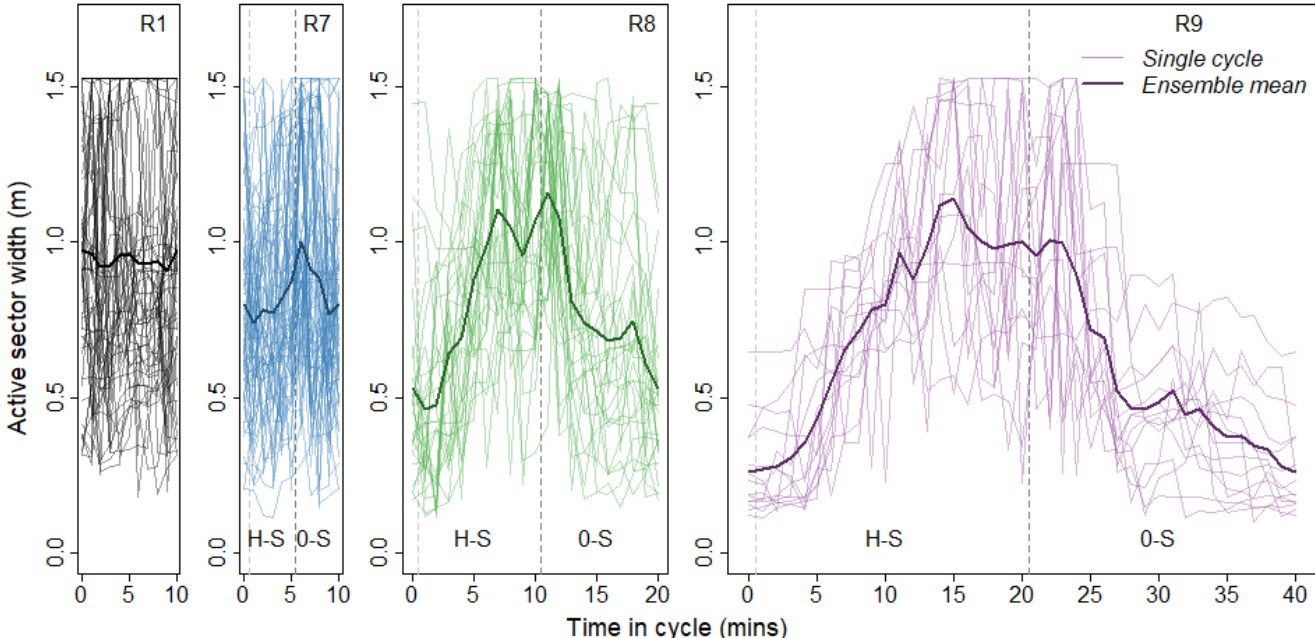

**Figure A4.** Width of active sector measured across **connected** channels at 1 m down-fan, versus time in experimental cycle. Thin lines represent each individual high-zero sediment supply cycle; thick line is the ensemble mean.



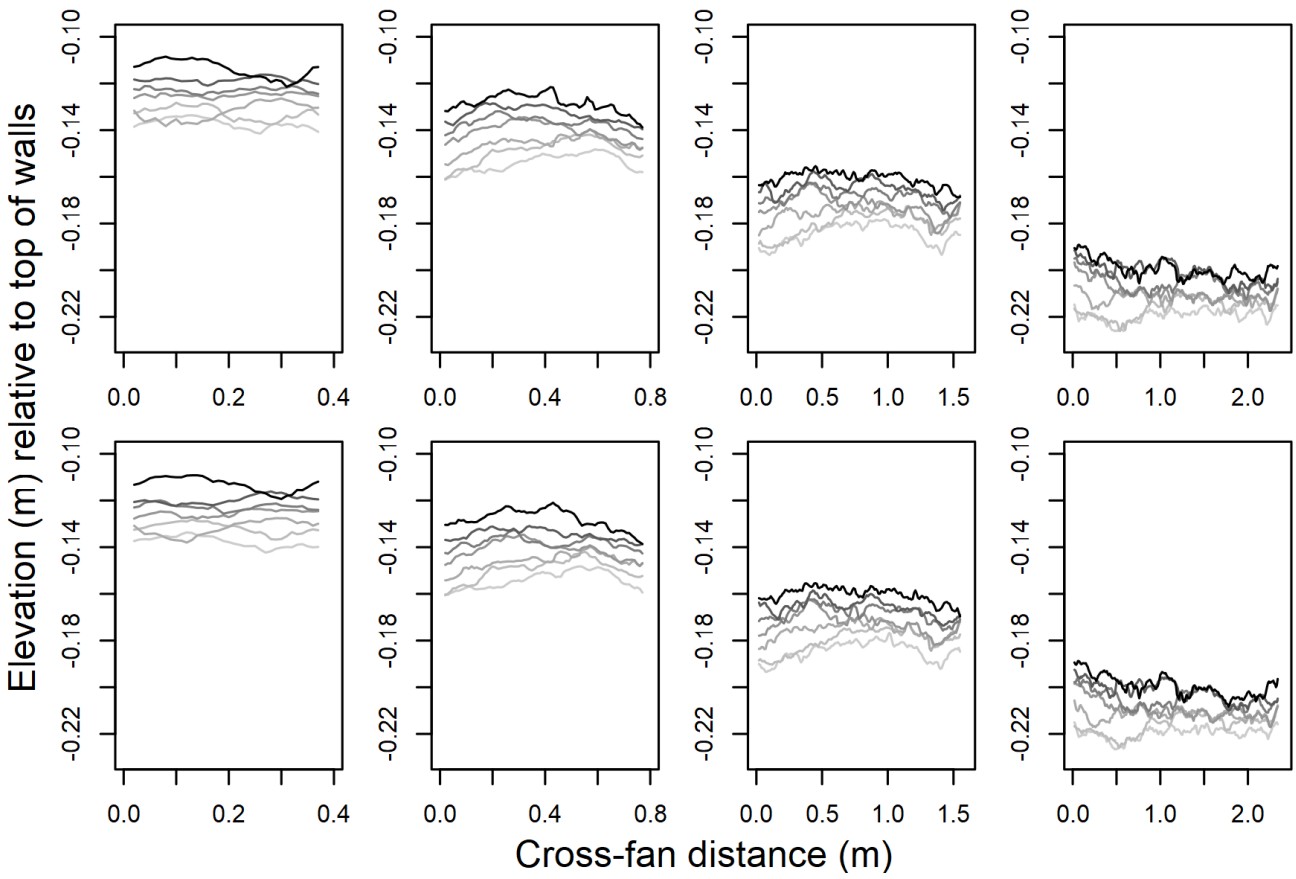

**Figure A5.** Fan elevation at arcuate cross-fan profiles in Run 7, located 0.25, 0.5, 1.0 and 1.5 m downfan (left to right). **Upper panel:** examples of fan topography at the end of zero-supply periods. **Lower panel:** examples of fan topography at the end of high-supply periods. Example data are from time steps 80 minutes apart; darker colours denote later time steps.



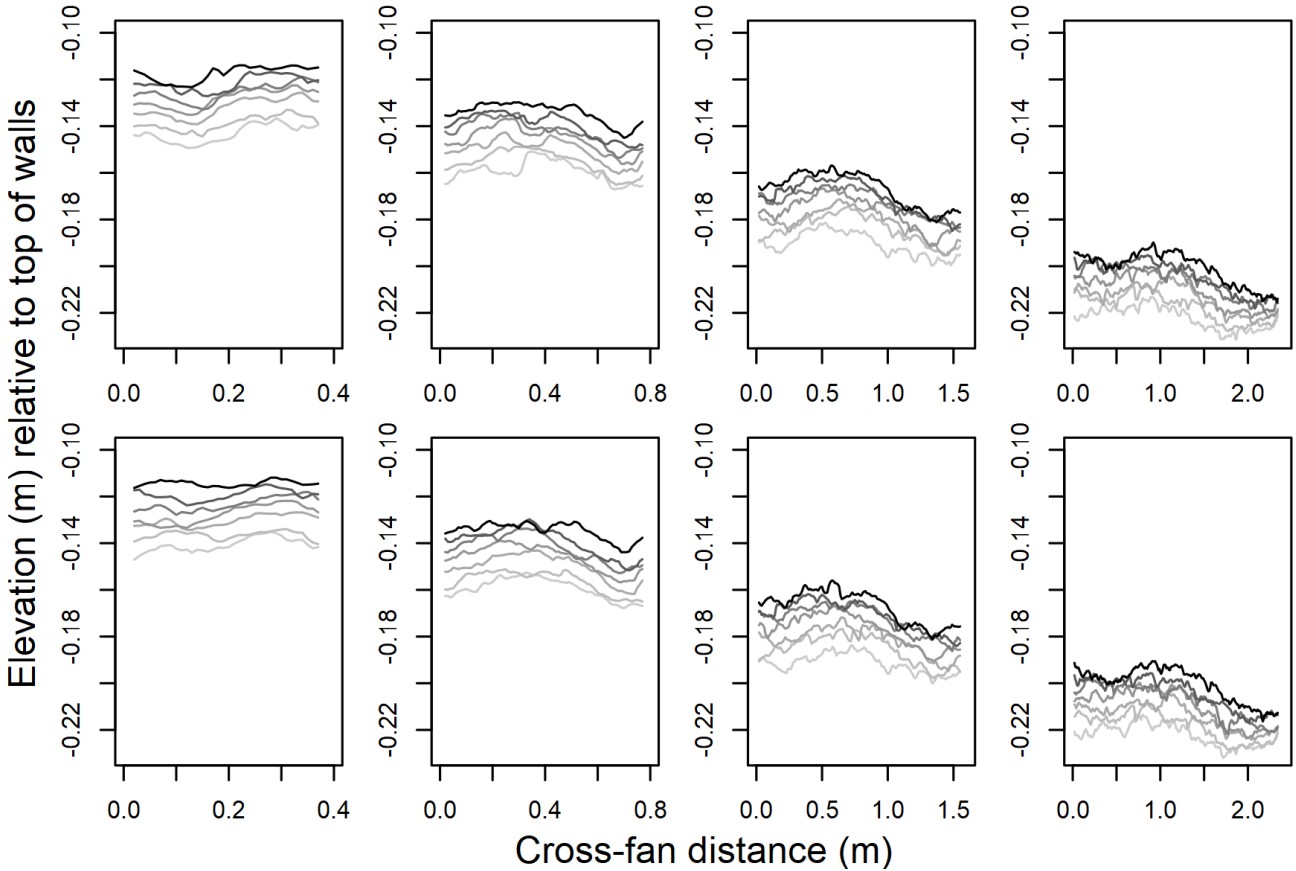

**Figure A6.** Fan elevation at arcuate cross-fan profiles in Run 8, located 0.25, 0.5, 1.0 and 1.5 m downfan (left to right). **Upper panel:** examples of fan topography at the end of zero-supply periods. **Lower panel:** examples of fan topography at the end of high-supply periods. Example data are from time steps 80 minutes apart; darker colours denote later time steps.



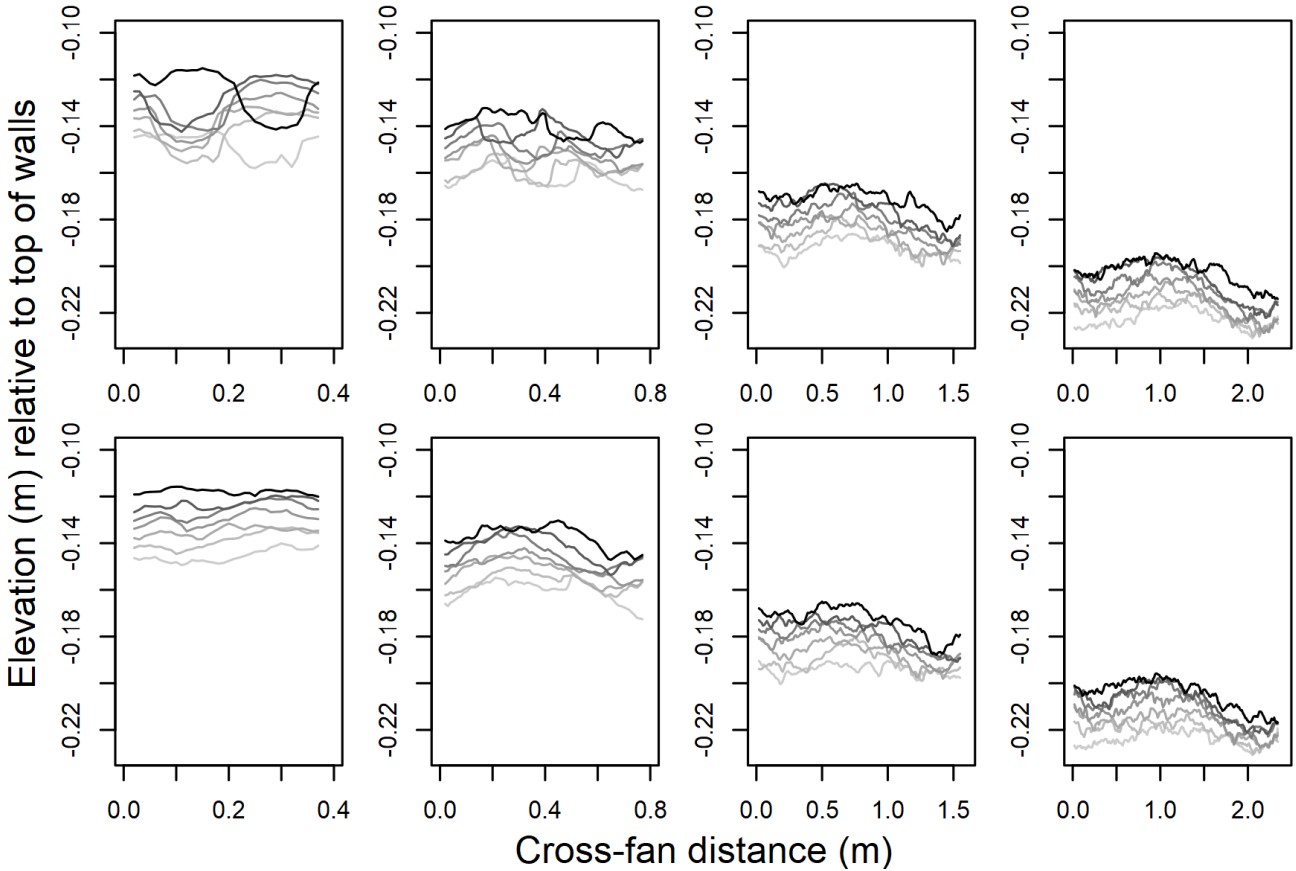

**Figure A7.** Fan elevation at arcuate cross-fan profiles in Run 9, located 0.25, 0.5, 1.0 and 1.5 m downfan (left to right). **Upper panel:** examples of fan topography at the end of zero-supply periods. **Lower panel:** examples of fan topography at the end of high-supply periods. Example data are from time steps 80 minutes apart; darker colours denote later time steps.





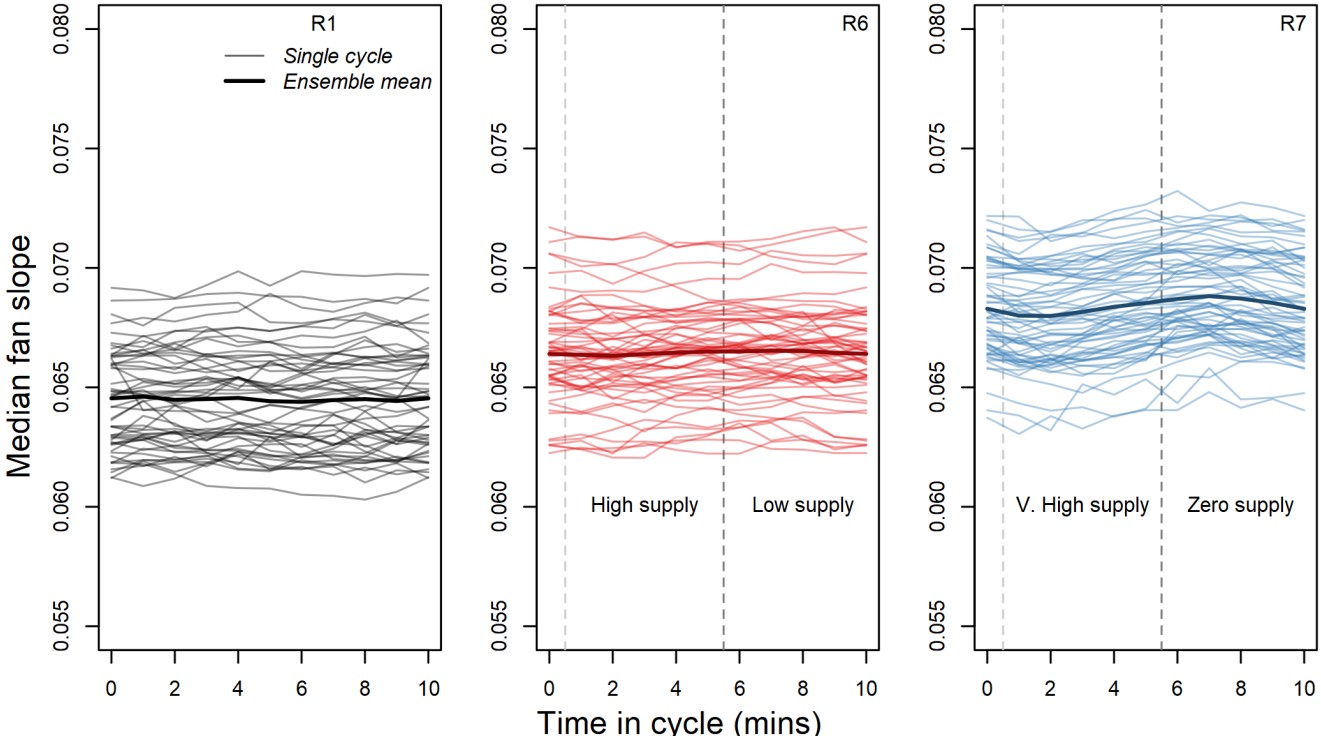

**Figure A8.** Median fan slope versus time in experimental cycle, for Runs 1, 6 and 7. Run 6 has the same mean sediment supply rate and timing of supply oscillations as Run 7, but has lower-amplitude oscillations. Thin lines represent each individual high-zero-supply cycle; thick line is the ensemble mean.





**Table A1.** The percentage of total sediment load or yield made up by bedload in mountain streams.

| Study | Location | % Bedload |
|---|---|---|
| Schoklitsch 1926 [1] | "Alpine Mountain Rivers" | 70 |
| Mcpherson 1971 | Two O'Clock Creek, Alberta, Canada | 0.5 |
| Bradley and Mears 1980 | Boulder Creek, Colorado, USA | 90 |
| Hayward 1980 | Torlesse Stream, Canterbury, NZ | 90 |
| Alvera and García-Ruiz 2000 | Izas catchment, Spain | 30 |
| Lenzi et al. 2003 | Rio Cordon, Italy | 24 |
| Métivier et al. 2004 | Ürümqi River (Chinese Tian Shan) | 45[2] |
| Meunier et al. 2006 | Torrent de St Pierre, France | 15-60[3] |
| Pratt-Sitaula et al. 2007 | Marsyandi River, Nepal | 36 |
| Alexandrov et al. 2009 | Nahal Eshtemoa, Israel | 5 |

[1] As cited in Jarocki (1957), p104.

[2] At the range front.

[3] Values quoted are minimum and maximum during a week of observations.

*Author contributions.* A. S. Leenman conceptualised the project, conducted the experiments, analysed the data, and wrote the manuscript. B. C.Eaton conceptualised the project, edited the manuscript, and provided supervision throughout the project.

*Competing interests.* The authors have no competing interests to declare.

*Acknowledgements.* A. S. Leenman was funded through a UBC Four-Year Fellowship. Experimental construction was funded through an NSERC Discovery Grant to B. C. Eaton. Thanks to Mike Church, Scott McDougall, and Sam Woor for lively discussions of fan dynamics and for helpful comments on an earlier draft.



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
