# Peer review of "Episodic sediment supply to alluvial fans: implications for fan incision and morphometry"

_EGUsphere, 2022_

## Author Response (AR1)

**Reviewer 1 comments and response:**

This paper presents the findings of four physical model experiments; this is a well established technique for evaluating processes on these landforms however the authors expand on previous research by using episodic sediment supply which is more representative of natural conditions during fan evolution. This provides a novel set of results which will be of interest to the ESurf readership both for those interested experimental methodologies, those in the alluvial fan community and also those with an interest in wider environmental reconstruction. The paper present a series of well created figures that display results and the links to the time-lapse videos of the experiments is a welcome addition, as is the provision of the model code. Overall I think that this paper should be accepted following some minor amendments detailed below.

**Specific comments**

The four experiments are labelled as Run 1, 7, 8 and 9 – I understand that this is due to results being presented in other papers and to maintain consistency but it is a confusing way to present the runs in this paper. It would be more appropriate to label them 1-4 or have an abbreviation to represent the degree of oscillation in the sediment supply, and in Line 50 refer to how they link to other publications and run numbers in these.

Relabelled with CON, OSC10, OSC20 and OSC40 following your second recommendation.

In Section 2.3 (Lines 109-111) you introduced the experimental scenarios that you used and mentioned that three repeats were carried for the constant sediment supply runs but only one set of data was presented in this paper, which was selected as it best fit the other runs. I think that you need a stronger justification here for why an average wasn't used and/or the implications of bias selection for this to showcase to the reader that you are presenting the unbiased results.

Instead of adding more justification, we have added data from the other two repeats of Run 1 (now named Run CON) to all relevant plots. The only plot to which we have not added data from repeats 1 and 3 is Figure 10; adding the extra data would smooth out the random oscillations in erosion/deposition across the fan that are visible in column 1 (Run CON), which provide important context for the strength of the signal in columns 2-4 (OSC runs).

In Section 5 (Conclusions) I think you can have a stronger final paragraph to emphasise the novelty and applicability of this work. I understand your point about your experiments being more realistic than previous constant supply experiments, however certainly in relation to my experiments (Clarke et al, 2010) they were never meant to represent long-term fan evolution but instead were highlighting that even

under constant climatic periods autogenic processes could still cause changing processes on alluvial fans. Therefore I think a stronger selling point of this work is that this represents the first study to use experiments to recreate alluvial fan processes that are more representative of natural conditions on alluvial fans, and therefore the processes shown are more directly applicable to hazard management and understanding the link between climate change (i.e. sediment-supply) and fan response, thus you are starting a shift-change in how alluvial fans should be modelled in experiments to start answering these real-World issues. Emphasising this at the end of the paper (and in the abstract) would broaden the appeal of the paper and showcase the strengths in your approach.

We have added to the final paragraph and abstract to both highlight the utility of constant input experiments, and to point out the utility of our own variable input experiments for understanding hazards, as you suggest (lines 11 and 488-490).

**Technical comments**

Line 34       At the end of the sentence "…most alluvial fan experiments use a constant sediment supply" it would be useful to cite the key papers here for – I know that you do refer to these later in the paper but they should be brought in here to show the body of previous work that you are building on.

Done.

Lines 43-47    I'm unclear why this is here. A summary of the key findings is provided in the abstract, discussion and conclusion and so doesn't need repeating here. Suggest removing.

We prefer to have a summary of key findings at the end of the intro to "signpost" what the reader should keep an eye out for while reading through the results – otherwise it is possible to get lost in the details of our data. However, we can remove this if the editors/other reviewers advise.

Lines 70-72    Repetition of start of Section 2.4 – remove from here and merge information into Section 2.4.

We feel this is worth keeping – here, it is to tell readers about the key data products from (i.e. what was the point of) our camera array. At the start of section 2.4, it is to reminder readers that these two products were the starting point for all subsequent analysis.

Line 89       Where on the fan was Re* estimated from? This would vary greatly depending on the position on the fan and so this needs clarifying as I was unclear whether this was calculated in the same position on the fan head as Fr or elsewhere.

Same position on the fan-head. Have edited line 105 for clarity.

Section 2.4      What were the error metrics for the SfM output and the DEM?

Detailed assessment in PhD dissertation (cited), with relevant information now cited in line 152-153.

Line 135        Unclear what you are trying to say here, suggest rephrasing.

Rephrased, now line 154-155.

Line 242        "Deposition rates then **decayed**…" – suggest using an alternative for decayed

Changed to "decreased"

Line 243        "…most evident in Runs 8 and 9" – this paragraph is focused on these two runs and so this is unnecessary but could be amended to Run 9 as this showed the most significant decline.

Opted to remove; you're right that it is unnecessary.

Lines 273/274 Suggest removing "that of" after the first and second points.

Rewrote these two sentences to remove "that of" and also be grammatically correct.

Line 290        "elongate" should be "elongate**d**"

Done.

Figure 9        Make sure that the ensemble lines for all of the plots are plotted in front of the individual oscillations so that these are clearly visible.

Done.

Figure A1:      Annotate to the aerial photograph to show the fan and catchment extents.

Done.

**Reviewer 2 comments and response:**

This paper presents results from experiments of alluvial fans to determine how abrupt changes in sediment supply affect fan morphodynamics and how the duration of sediment supply oscillations affects fan response. The authors present a concise and well-written story that examines several features and behaviors from a set of four experiments, showing that time-varying sediment supply causes changes in large-scale fan morphology and channel dynamics, and that increasing the period of the oscillations creates more elongate fans with lower slopes. This paper is an important contribution to the experimental and alluvial fan communities, and provides a sound argument for changing how we model these systems. I suggest the paper be published after some minor revisions.

**Comments:**

The introduction would benefit from a brief discussion of what we know about adjustment timescales relative to event timescales in alluvial fans. I know this is touched on in the discussion, but this is a classic concept in geomorphology and is worth highlighting. You are looking to determine if changes in sediment supply over time will have any impact on the system, so this will clearly be related to the timescale of the change relative to the timescale of system adjustment or an avulsion timescale. This would set you up to discuss your results in the context of timescales later on, which I also suggest placing more emphasis on in the discussion, as it would be helpful to explain a lot of your results.

A useful suggestion. Added a paragraph to this end in the introduction (lines 30-40). Have also discussed timescales more explicitly in the discussion (lines 368-372).

Why were there 3 (or 4?) experiments done for Run 1 but only 2 for Runs 7-9? Further, why did you do repeats but then only present results from one experiment? It would be beneficial to show the results averaged from all of the experiments, not just one.

There were 3 exps for Run 1 (CON) and only 1 for Runs 7-9 (OSC); have clarified this (line 127-129). The main reason for this is that I ran out of time to conduct additional repeats due to lab renovations and then Covid-19. Future replicates of these experiments would be useful and interesting and I'd love to run them when I get another fan-simulator up and running. The student currently using the UBC fan simulator is not conducting additional repeats of these experiments as she is taking her PhD in a different direction. I have not added this lengthy explanation to the paper (it seems a little superfluous) but can do so if the editors advise.

R.e. Showing the results from all experiments: have added data from all repeats of Run /OSC1 to the relevant figures (except Figure 10, where doing so would inhibit comparison between runs).

You have selected particular distances at which to measure things in your data analysis (e.g., 0.25 m down-fan for fan head entrenchment). Please provide a justification for selecting these distances.

Added detail to the methods to explain our choices (lines 161 and 167).

Figure 5, and all figures of a similar nature, would benefit from including text on the figures that lists the mean or median value for each experiment. On Figure 5, for example, we can see the difference in how slope changes over time, but what is the time-averaged slope for Run 1 compared to Runs 7-9? Please include this informataion on each figure. If you're also going to talk about variance, as you do later, put that number on the figure along with the mean/median.

Mean and standard deviation added to all plots where relevant.

Figure 8: What is the physical reasoning for the spike in Fn after sediment supply is turned off? Is it related to excavation of the feeder channel? Some discussion of this would be appreciated, as it is not necessarily expected or intuitive.

It's because the channel is actively reshaping itself from multi-thread to single-thread (often a new, central single-thread channel), so mobility is temporarily very high. We had implied this but have now explained the physical mechanism more clearly (now lines 248-250).

Looking at Figure 5 and Table 1, I suspected that if the channels become more entrenched with the longer periods, those entrenched channels would have less or slower lateral migration. There may also be a larger difference between the channel gradient and fan gradient for fans with longer periods of oscillation (Run 9) compared to those with shorter periods (Run 7). Combining this with Figure 8, by eye, it looks like there is overall faster migration in Run 7 compared to Run 9, possibly due to this entrenchment. What does this tell us? There could be drastically different adjustment times, avulsion times, and overall system dynamics if the channel gradients are very different from fan gradients.

The average migration rate is actually very similar for runs 7 and 8, and slightly lower for Run 9. What really stands out is that the lowest migration rates are right at the end of the zero-supply period. As you say, this acts to enhance the efficiency of down-cutting because flow is concentrated in one place at the fan-head. Have added text to this effect in the discussion of trenching (lines 365-367).

Figure 9: Can these be separated out into separate plots? it's really quite difficult to see anything. Or at least have a row of erosion, a row of deposition, and then a row where they're combined like this

The idea was to show the relative magnitudes of erosion and deposition, particularly at the end of Run 9/OSC40. Nevertheless, I concur that the data are difficult to see (especially the mean deposition, as Dr Clarke also notes). Have remade as two rows of plots, following your suggestion (colours now match previous plots as well). The original plot is still retained in the SM if readers want to compare by eye.

*It would be helpful to show fraction area abandoned alongside fraction area newly inundated to observe the change in channel network structure as it reorganizes.*

The channel network structure can best be gleaned from Figure 6 (the number of channel threads) and Figure A3 (same plot, farther up-fan). Figure A4 is also useful, as it shows the sector of the fan that is spanned by active flow (implying diverging or channelised flow with higher and lower values, respectively).

Adding the fraction abandoned certainly would help elucidate the timing of mobility peaks. However, I'm not sure the extra data would be as helpful as that already included in Figures 7 and 8. I don't want to overwhelm the reader with information, but could definitely add the fraction abandoned (perhaps as a separate panel for Figure 8) if the editors advise.

*Regarding fan head trenching (discussion lines ~319-326), how do you reconcile the fact that previous studies did get a trench with constant Qs? You have replicates, so is it true across all replicates that you did not get trenching in Runs 1 and 7, only 8 and 9? Later, your explanation for why there is a trench in 8-9 but not 7 doesn't quite get there. You suggest a decrease in the number of channels during 0-S that concentrates flow and erosion in a few channels, leading to downcutting. But you have periods of low sed supply in all runs, so it must be related to some response or adjustment timescale, which in Run 7 is likely longer than the period of oscillation, no?*

The lack of trench development in Run 1/CON relates to the highly dynamic nature of the experimental channels – they simply didn't stay in place long enough to incise a trench! In a previous paper we hypothesise that this reflects the generally high sediment concentration and wide GSD in our experiments, both of which facilitated channel sedimentation. Have added a citation where relevant (lines 355-356).

It is true across all repeats that no persistent trench developed for Run 1/CON. See for instance Figure 6, which now contains data for all three repeats of Run 1/CON. There were no additional replicates for Runs 7-9/OSC (this fact has also been clarified now).

Yes, the lack of trench in Run 7 and presence in Runs 8-9 is absolutely related to an adjustment timescale, and to the duration of the zero-supply period relative to that adjustment timescale. This was implicit in lines 327-334 (old version) but as you say,

the explanation didn't quite get there. Have added text to fully explain what's going on here, and to reflect more upon what this tells us about disturbance vs adjustment timescales in our experiment (lines 362-368).

**Technical comments:**

Table 1 should include the standard deviation and error.

Standard deviation added to table; error added to caption.

L151: 7 mm is not sand. Rephrase to just say "approximately the size of the largest grains."

Done.

You note that video speed differs because photos were collected at different intervals. This can be corrected when the videos are made by adjusting the frame rate. Please consider adjusting this so readers can see the difference in relative rates of channel mobility.

This difference only affects the comparison between Run 1/CON and the other runs. We feel that the most interesting comparisons are between videos of the OSC runs, which are all at the same speed. However, we're happy to remake the video for Run 1/CON with a slower frame-rate to match the OSC videos if the editors agree that it would be helpful.

L163: "for their self-organised adjustment **the** input conditions or changes thereof" change 'the' to 'to'

Added 'to'.

It would be helpful to remind the reader in the figure captions and/or text if these various metrics you're plotting are at a transect or fan-averaged.

Done.

L314: The experiments don't say anything about amplitude, only period. You have the one prelim experiment in supplemental, but I'd focus the discussion here on duration for what you can actually show.

We've kept this sentence as we felt that the reference to our first low-amplitude experiment was useful to highlight how dampened variations had a lesser effect. This is the only place where we comment on amplitude; the rest of our discussion considers duration alone.

---

## Author Response (AR2)

Dear Dr Clubb,

Thank you for taking the time to look over our revisions, as well as your further suggestions (which are really helpful).

Thanks for catching the mislabelling of the QR codes - the rasterized labels managed to escape my 'replace all'. Have re-labelled the QR codes and included snapshots of the fan as well, which is another good idea; thank you.

I too had hesitations about the longevity of QR codes and youtube links, but I do appreciate the accessibility of these tools for today's users. For future proofing, I've added the videos to the Zenodo repository as you suggest. Have updated the DOI for the new Zenodo repo version in the "data availability section".

That's a good idea r.e. reifying the code in a tagged version. Have created a tagged release of each GitHub repo and loaded the zipped repositories into the same Zenodo repository. I'll keep the original GitHub links as well – I think it's a little easier to download the tagged versions from there.

R.e. Figure 9 old vs new: I really liked how easy it was to compare erosion and deposition too! I think the figure will be too unwieldy with a third row of subplots though, so I've gone with your suggestion of adding the mean of each erosion subplot to the corresponding deposition plot and vice versa. Let me know what you think.

Please let me know if you have any further comments or questions.

With best wishes,

Anya